# Chronic nicotine increases midbrain dopamine neuron activity and biases individual strategies towards reduced exploration in mice

Malou Dongelmans[1,5], Romain Durand-de Cuttoli [1,4,5], Claire Nguyen [1,5], Maxime Come [1,2,5], Etienne K. Duranté [1], Damien Lemoine[1], Raphaël Brito[1], Tarek Ahmed Yahia[1], Sarah Mondoloni [1], Steve Didienne[1,2], Elise Bousseyrol [1,2], Bernadette Hannesse[1], Lauren M. Reynolds [1,2], Nicolas Torquet[1], Deniz Dalkara [3], Fabio Marti [1,2], Alexandre Mourot [1,2], Jérémie Naudé [1,2] & Philippe Faure [1,2✉]

Long-term exposure to nicotine alters brain circuits and induces profound changes in decision-making strategies, affecting behaviors both related and unrelated to drug seeking and consumption. Using an intracranial self-stimulation reward-based foraging task, we investigated in mice the impact of chronic nicotine on midbrain dopamine neuron activity and its consequence on the trade-off between exploitation and exploration. Model-based and archetypal analysis revealed substantial inter-individual variability in decision-making strategies, with mice passively exposed to nicotine shifting toward a more exploitative profile compared to non-exposed animals. We then mimicked the effect of chronic nicotine on the tonic activity of dopamine neurons using optogenetics, and found that photo-stimulated mice adopted a behavioral phenotype similar to that of mice exposed to chronic nicotine. Our results reveal a key role of tonic midbrain dopamine in the exploration/exploitation trade-off and highlight a potential mechanism by which nicotine affects the exploration/exploitation balance and decision-making.

[1] Sorbonne Université, INSERM, CNRS, Neuroscience Paris Seine - Institut de Biologie Paris Seine (NPS - IBPS), 75005 Paris, France. [2] Brain Plasticity Unit, CNRS, ESPCI Paris, PSL Research University, 75005 Paris, France. [3] Sorbonne Université, INSERM, CNRS, Institut de la Vision, Paris, France. [4] Present address: Nash Family Department of Neuroscience, Icahn School of Medicine at Mount Sinai, New York, NY, USA. [5] These authors contributed equally: Malou Dongelmans, Romain Durand-de Cuttoli, Claire Nguyen, Maxime Come. ✉email: phfaure@gmail.com

Nicotine is the primary reinforcing component driving tobacco addiction[1–3]. Like most addictive substances, nicotine is hypothesized to perpetuate addiction through alterations in dopamine (DA) signaling and plasticity in the mesocorticolimbic pathway[4]. Repeated activation of ventral tegmental area (VTA) DA neurons by nicotine not only leads to reinforcement but also to craving and lack of self-control over intake[5]. Concurrently, chronic exposure to nicotine exerts numerous effects on brain and circuits, affecting personality traits and behaviors that extend beyond drug-seeking or consumption,[6,7] such as changes to emotional state or levels of stress[8,9] and anxiety[10]. Chronic nicotine exposure also impacts various components of decision-making processes, such as impulsivity[11,12] or exploratory behaviors[13,14] which may contribute to the persistence of drug consumption by promoting relapse and susceptibility to other addictions[15]. However, directly linking the cellular effects of nicotine to modifications of decision-making has been elusive. Understanding the molecular and circuit-level mechanisms of nicotine on decision-making is needed to decipher its multifaceted effects. Here we take advantage of a decision-making framework in a rodent model to address the impact of chronic nicotine exposure on VTA DA neuron activity and decision-making parameters.

Among the components of decision-making, the explore/exploit trade-off is of particular interest. Exploitation refers to choosing the option that seems, based on the history of rewards, the optimal choice. However, when faced with two alternatives, one with low and one with high probability of reward, animals do not purely exploit, they also choose the less likely rewarded option a significant portion of the time. The origin of such seemingly suboptimal choices remains poorly understood. It has been interpreted as noise, error, risk-seeking, irrational belief, or exploration[7,16–19]. In the context of exploration, choosing an option with less likelihood of immediate reward is essential to gather information about unknown or uncertain outcomes in a changing environment. As new information is crucial for learning and behavioral adaptation[7,17], exploration is central to the emergence and organization of behaviors[20]. Nevertheless, optimizing behavioral strategies require to exploit reward knowledge. Exploitation and exploration thus constitute important, yet opposing, adaptive processes. Hence, determining the exact trade-off between exploration and exploitation is key to decision-making. This trade-off is ubiquitous across species and pervades a number of altered behaviors under specific psychiatric conditions, such as addiction[6,7]. It is thus an ecologically valid tool for translational research and for dissecting the link between the impact of the drug at the molecular, circuit and behavioral levels. In the context of addiction, a modification of this trade off will impact the global equilibrium of decisions between drug and non-drug rewards. Determining whether chronic nicotine exposure alters such exploration–exploitation trade-off is thus fundamental to help understand modifications of individual traits associated with continued nicotine consumption.

Altered DA function is a promising candidate to link chronic nicotine exposure to changes in decision-making behavior. This neuromodulator, which is at the crossroads of motivation, learning and decision-making, is hijacked, in the context of addiction, by most drugs of abuse[21–23]. Changes in the spontaneous tonic firing of VTA DA neurons, as a consequence of repetitive drug use, can indeed alter the subjective value assigned to available rewards[21], as well as the motivational salience of the drug or of drug-predicting cues[24], influencing decisions about which reward to pursue[25]. Tonic DA can scale the performance of a learned behavior[26], the incentive value associated with environmental stimuli[27], or signal the average reward[28]. In the exploration/exploitation framework, the role of tonic DA remains debated. The effect of DA manipulation on the exploration/exploitation balance is convincing but varies depending on the task[29–31]. Increasing tonic striatal DA release has been suggested to either increase[29] or decrease[31] the level of exploration. Decreasing tonic striatal DA has also been suggested to increase exploration[32]. Hence, drug-induced alterations of DA transmission may modify behavioral choices, either positively or negatively depending on the environment and the specific type of DA manipulation.

We have shown that decisions in reward-based foraging are modulated by the cholinergic neurotransmission of the VTA, with a particular role of nicotinic acetylcholine receptors in directed exploration, driven by expected uncertainty[33]. Here we demonstrated that chronic nicotine exposure increases the tonic activity of VTA DA neurons and reduces undirected exploration to favor exploitation, with mice focusing on the most valuable options at the expense of information gathering. Acutely increasing the tonic activity of VTA DA neurons using optogenetics is sufficient to mimic the behavioral bias (decreased exploration) induced by nicotine, suggesting that the DA control of the exploration/exploitation balance is altered by long-term nicotine exposure.

## Results

**Mouse choices depend on reward probability, uncertainty and on motor cost.** To assess choice behavior in an uncertain environment, we used a multi-armed ICSS (intracranial self-stimulation) bandit task for mice where specific locations, hereafter called targets, were associated with medial forebrain bundle (MFB) stimulations as rewards (Fig. 1a and Supplementary Fig. 1)[19,33]. The task takes place in a circular open-field (interior diameter = 68 cm), with three explicitly marked targets forming the apices of a triangle (Fig. 1b). Passing over each target results in the delivery of a rewarding intracranial electrical stimulation. Mice cannot receive two consecutive stimulations at the same target, and thus learn to forage from one target to another in order to continue receiving stimulations (Fig. 1b, left). During the training period (5 min daily sessions), hereafter called the deterministic setting (DS, Fig. 1c, left), every visit to a target was reinforced by a stimulation (reward probability $P = 100\%$ at each location, $P_{100}$). At the end of the DS, mice were confronted with a probabilistic setting (PS, Fig. 1c, right) where each target was now associated with a different probability of stimulation delivery ($P = 100\%$, 50%, and 25%, Fig. 1c, right). As previously shown[33], the PS induced a marked change in the behavioral pattern compared to the deterministic one. Trajectories at the end of the DS were almost circular, with very few directional changes (i.e., returning to the previous target, Fig. 1d) due to the associated motor cost (mice have to do a U-turn instead of going forward)[19]. In contrast, mice distributed their choices differently in the PS by incorporating more directional changes—an adaptation from the circular strategy (Fig. 1d). Directional changes in the PS were not random: rather, they allowed animals to focus on specific targets. Indeed, compared to the DS where mice visited the three targets with a uniform distribution, in the PS mice visited more often the targets associated with the highest reward probabilities (i.e., $P_{100}$ and $P_{50}$, Fig. 1e). Contrary to a purely exploitative strategy with alternating visits between $p_{100}$ and $p_{50}$, mice continued to visit all three points, prompting us to further investigate the exploration/exploitation trade-off in their choices. However, the global repartition of visits does not directly measure choices. Indeed, since mice cannot receive two consecutive rewards from the same target, the repartition of visits on targets is the result of binary choices in three gambles ($G_{100}$, $G_{25}$, $G_{50}$) between two respective payoffs (here, $G_{100} = \{P_{50}$ vs $P_{25}\}$, $G_{25} = \{P_{100}$ vs $P_{50}\}$, $G_{50} = \{P_{100}$

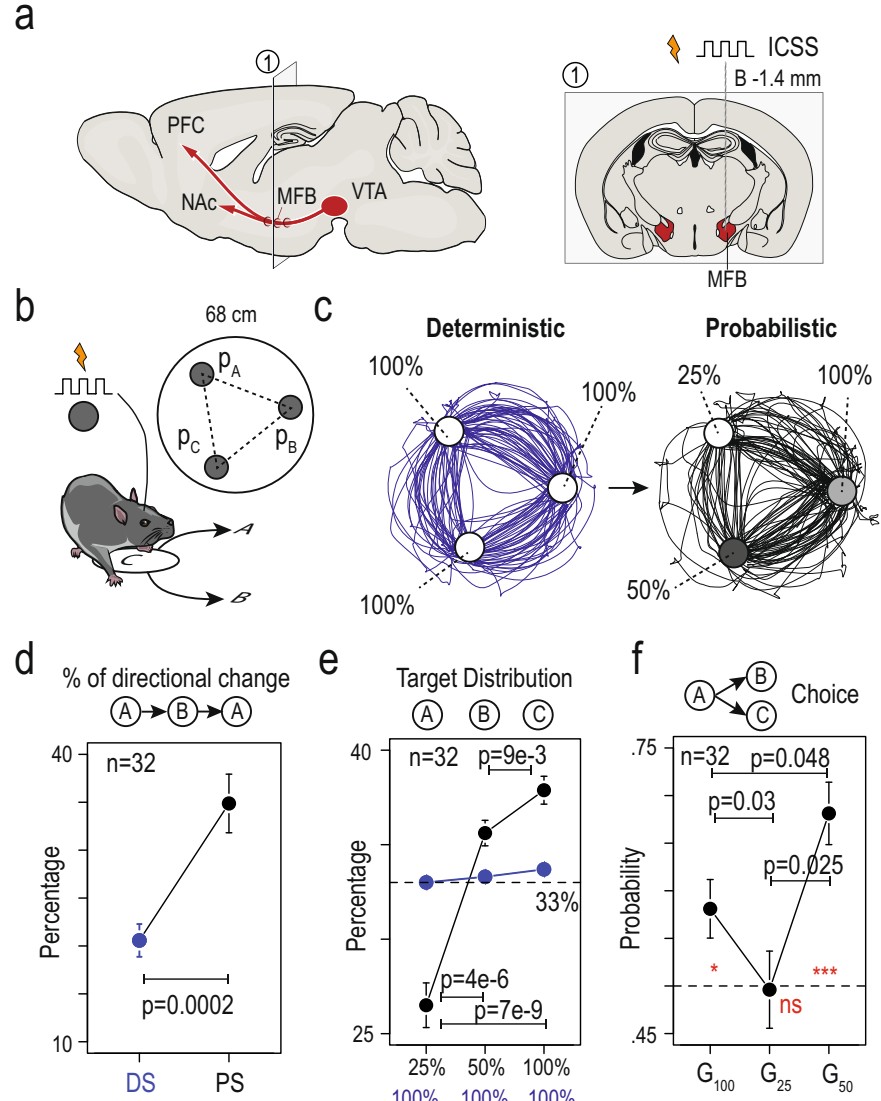

**Fig. 1 Mice exhibited suboptimal behavior and exploratory choices in a spatial version of a multi-armed bandit task with probabilistic settings. a** Mice were implanted unilaterally with bipolar stimulation electrodes to deliver electrical stimulation at the level of the medial forebrain bundle in order to support intracranial self-stimulation (ICSS) behavior. Right: a coronal section of the mouse brain illustrating a representative electrode positioned in the MFB at bregma −1.4 mm AP. **b** Schematic of the behavioral paradigm: mice are placed in a circular open-field (interior diameter = 68 cm), with three equidistant targets (A, B, and C, labeled on the open-field floor, 35 cm distance between targets) that are associated with a given probability ($P_A$, $P_B$, or $P_C$) of ICSS reward delivery when the animal is detected in a 60 mm zone around the target. **c** Sample trajectories for one mouse under the deterministic setting (DS) of the task, in which each of the three targets were rewarded by an ICSS with $P = 100\%$ (left panel, blue), and in the probabilistic setting (PS), in which the three targets were associated with distinct probabilities of ICSS delivery ($P_A = 100$, $P_B = 50$, and $P_C = 25\%$) (right panel, black). Two stimulations could not be delivered consecutively in the same zone, therefore animals learned to alternate between targets with a circular pattern in the DS (blue), and a less stereotyped pattern in the PS (black). **d** Comparison of the percentage of directional changes during DS (blue) and PS (black) (two-sided paired Wilcoxon rank test, $p = 0.0002$, $n = 32$). **e** Repartition of visits to the three targets. Under the DS (blue), animals distributed uniformly their choices of visiting each of the three options (around 33%, Friedman rank sum test, $p = 0.82$). During the PS (black), animals reorganized their behavior and visited more frequently options with greater probabilities of reward (Friedman rank sum test, $p = 7e-8$, and two-sided paired Wilcoxon Test with holm correction $p = 4.e-6$, 7e-9 and 9e-3 for the three comparisons, $n = 32$). **f** Probability to choose the option with the highest probability of reward for the three possible gambles: $G_{100}$ = choice of 50% over 25%, $G_{25}$ = choice of 100% over 50%, and $G_{50}$ = choice of 100% over 25%. Red asterisk: Comparison with a true mean of 0.5 (two-sided one sample $t$-test with Holm correction, $n = 32$) for $G_{100}$ ($p = 0.026$), $G_{25}$ ($p = 0.92$), and $G_{50}$ ($p = 1.4e-5$). Paired comparison (paired $t$-test with Holm correction, $n = 32$) for $G_{100}$–$G_{25}$ ($p = 0.03$), $G_{100}$–$G_{50}$ ($p = 0.048$), and $G_{25}$–$G_{50}$ ($p = 0.025$). For (**d**–**f**), data are shown as mean ± sem. Boxplot shown median, quartiles, and extreme values. See also source data.

vs $P_{25}$}) (Fig. 1f). Hence, we investigated free choices in each gamble independently, and the resulting trade-off between exploitation and exploration. When faced with a choice between two alternatives, exploitation corresponds to choosing the option for which the animal assigns the highest value, while exploration

corresponds to choosing the less valued alternative. Animals purely exploiting would always choose the high-probability option, but we found that mice chose the less likely rewarded option a significant portion of the time, consistent with balancing exploitation and exploration in their choice behavior. For $G_{100}$

and $G_{50}$, mice chose the optimal location (i.e., the one associated with the highest probability of reward) in more than 50% of trials. However, for $G_{25}$ (i.e., the free binary choice between $P_{100}$ and $P_{50}$ when the animal is on $P_{25}$) the probability to choose $P_{100}$ over $P_{50}$ was not different from a random choice (Fig. 1f), which we interpreted as mice assigning a positive motivational value to expected uncertainty, which is maximal at $p_{50}$[33]. Overall, mice biased their choices depending on the motor cost, and the probability and uncertainty of reward delivery. Behavior in the task was therefore the result of a balance between exploratory and exploitative choices.

**Nicotine exposure biases choices toward the most valuable options and promotes exploitation**. We next aimed to investigate the effects of chronic nicotine exposure on decision-making behavior and on the balance between exploration and exploitation in the same task. To do so, we implanted osmotic minipumps subcutaneously to expose mice to continuous nicotine (Nic, 10 mg/kg/day) or saline (Sal) for 3 weeks and then compared their behavior at the end of the DS and in the PS of the ICSS task (Fig. 2a). Because nicotine induces long lasting adaptations in the midbrain DA system[34], and because VTA DA neurons have been associated with decision-making under uncertainty[22,33], we first analyzed the spontaneous tonic activity of VTA DA cells in anesthetized mice. We recorded neurons from mice chronically exposed to either saline or nicotine via minipump, and that either had performed the behavioral task ("ICSS", at the end of PS), or were behaviorally naive. DA neurons firing was analyzed with respect to the average firing frequency and the percentage of spikes within bursts (%SWB). As previously reported[9,35], chronic exposure to nicotine increased the tonic activity of DA neurons, both in terms of firing frequency and bursting activity, when compared to mice implanted with a saline minipump, in both mice that performed (ICSS) or not (no ICSS) the task (Fig. 2b). Furthermore, mice exposed to the ICSS task exhibited an increase in firing frequency, but no change in bursting activity when compared to mice that were not stimulated (Fig. 2b).

We then analyzed the behavior of mice in the ICSS task. Overall, we did not see any behavioral difference between mice implanted with a saline minipump ($n = 23$) and the non-implanted mice ($n = 32$) analyzed in Fig. 1 (Supplementary Fig. 2a–c). Therefore, these two groups were pooled and henceforth referred to as control (Ctl, $n = 55$). Trajectories at the end of the DS were stereotyped, almost circular, in both control and nicotine-treated mice. Both groups distributed their visits equally over the three locations (Fig. 2c) and their respective probabilities of directional changes were equal ($\Delta = -2.7\%$, Fig. 2d). However, the total number of rewards was higher for nicotine-treated than for control mice ($\Delta = 26$, Fig. 2e), as a consequence of the decrease in the mean time-to-goal (i.e., the time necessary to go from one target to the next) in nicotine-treated mice ($\Delta = 0.83$ s, Fig. 2f). When mice were placed in a classical open-field (without ICSS), a greater velocity was observed in mice exposed to nicotine, yet only at the beginning of the session (first 5 min) (Supplementary Fig. 2d). This result suggests that the increased speed observed in the ICSS task for nicotine-treated mice may arise from the combined effects of nicotine exposure and the stimulation rewards.

Clear differences in the behavior of nicotine- and saline-exposed mice were observed in the PS. Both groups distributed their choices depending on the probability to receive a reward, but with markedly different strategies. Notably, while control mice visited significantly $P_{25}$, nicotine-treated mice instead focused on visiting the two most rewarded options (i.e., $P_{50}$ and $P_{100}$, Fig. 2g, $\Delta_{25} = -5\%$, $\Delta_{50} = 2.7\%$, $\Delta_{100} = 2.3\%$), which was

associated with an increase in the percentage of directional changes ($\Delta = 11\%$, Fig. 2h). These alterations in overall repartition resulted from changes in successive binary choices, with an increase in optimal choice selection in gamble $G_{100}$ (Fig. 2i, $\Delta = 10\%$) for nicotine-treated mice compared to control mice. We also observed an increase in the total number of obtained rewards ($\Delta = 17.9$, $p = 0.002$) and in the percentage of success (number of rewards divided by the number of trials, $\Delta = 2\%$, $p = 0.02$) in nicotine-treated mice compared to control mice. Finally, the comparison of mean time-to-goal between the two groups ($\Delta = -1.1$ s, Fig. 2j) indicates again an increased velocity in nicotine-treated mice, as was already observed in the DS. This increase in speed in the PS was not associated with a decrease in the number of directional changes made by nicotine-treated mice, suggesting that animals did not enter an automatic circular mode, disengaged from actual choices, but instead remained in a deliberative process. Altogether, these results indicate that chronic nicotine modifies the decision-making strategy of mice by biasing choices towards the seemingly most valuable options.

In the PS, adopting a purely exploitative strategy to maximize the success rate would require mice to choose the alternative with the highest probability of reward in each gamble, leading to a sequence of choices with solely the alternation of visits between $P_{100}$ and $P_{50}$. Both control and nicotine-treated groups clearly deviated from this strategy of pure exploitation, although nicotine-treated mice were more exploitative on average. Yet, population analyses (i.e., averaging over groups of animals) classically do not reflect the wide range of distinct behaviors and strategies that can be adopted by individuals. We therefore further analyzed our behavioral data, with the aim of revealing individual profiles and their adaptation under nicotine exposure.

**Idiosyncrasy in choice behavior suggests individual strategies**. Visual inspection of individual trajectories revealed that in the PS, some mice retained a circular strategy (with either an ascending $\{P_{25} - P_{50} - P_{100}\}$ or descending $\{P_{100} - P_{50} - P_{25}\}$ order) while others had what we hereafter call a gain-optimizing (GO) strategy, alternating between targets associated with the highest reward probabilities ($P_{100}$ and $P_{50}$) (Fig. 3a, lower left). Through "gain-optimizing" strategy, we mean a very basic definition of optimality based only on maximizing the number of rewards, but which does not take into account the potential advantage of exploration. Theoretically, always choosing the most valuable option would lead to an average success rate of 75% (Fig. 3a, lower right) while a purely circular strategy would lead to an average estimate of 58.3% success rate (Fig. 3a, upper right). Accordingly, the percentage of directional changes was correlated with the success rate (Fig. 3b, for control and nicotine-treated mice). In this graphical representation, the line (Fig. 3b, red line) that connects the theoretical points of the circular strategy (0% directional change, 58.3% success) and of the GO strategy (100% directional changes, 75% success) represents a progressive shift in strategy. We found that, experimentally, the slope ($s = 17.1 \pm 1.5$, black line, Fig. 3b) of the correlation between the percentage of directional changes and success rate was almost parallel to the theoretical line from circular to GO strategies ($S_{th} = 16.7$, red line, Fig. 3b), indicating that most of the directional changes were not random, but consisted in back-and-forth sequences between the $p_{50}$ and $p_{100}$ targets.

Differences in individual choice patterns were neither due to random variations, nor to different learning speeds, but rather a consequence of robust individual strategies. This is suggested first by the overall stability of the behaviors as indicated by the convergence to a plateau at sessions 8–10 (Supplementary Fig. 4a), and by the absence of any positive correlations between decision-

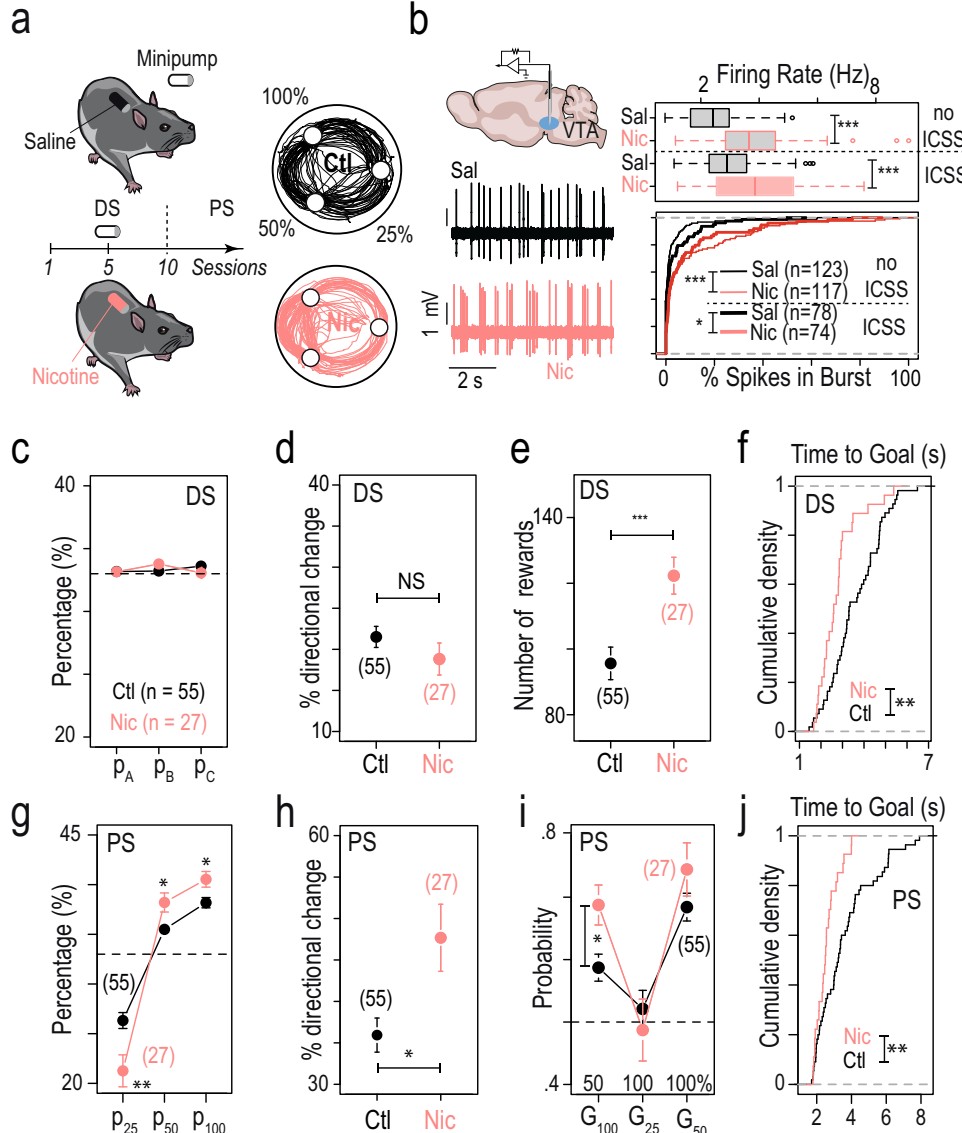

**Fig. 2 Chronic exposure to nicotine altered both spontaneous DA activity and choice strategies. a** Left: timeline of the task. Subcutaneous osmotic minipumps delivering nicotine (10 mg/kg/day), or saline for control animals, were implanted on day 5 of the deterministic setting (DS). Right: sample trajectories at the end of the probabilistic setting (PS) for a mouse under chronic nicotine (Nic, in pink) and a mouse naive to nicotine (Ctl, in black). **b** Left: representative in vivo electrophysiological recordings of VTA DA neurons after chronic saline (Sal, black) or nicotine (Nic, red) exposure. Right: the firing frequency and bursting activity of VTA DA neurons were compared between two sets of conditions: saline ($n = 123$ neurons) vs nicotine minipump ($n = 117$), and saline minipump + ICSS ($n = 78$) vs nicotine minipump + ICSS ($n = 74$) after completion of the PS. All electrophysiological experiments were performed after $24 \pm 2$ days of Sal or Nic (10 mg/kg/day) exposure. Nicotine exposure increased both DA neuron firing frequency (two-way ANOVA, nicotine effect $F_{(1,388)} = 77.57$, $p < 2e\text{-}16$) and bursting activity ($F_{(1,388)} = 25.14$, $p = 8.1e\text{-}7$), with or without ICSS. This increase was observed between the Sal and Nic minipump-only conditions (post hoc two-sided Wilcoxon test with Holm correction for multiple comparisons, firing frequency $p = 2.2e\text{-}10$, bursting activity $p = 1.3e\text{-}7$), as well as after Nic minipump + ICSS compared to Sal minipump + ICSS (firing frequency $p = 6.4e\text{-}4$, bursting activity $p = 0.014$). Mean firing frequency was increased after ICSS in both the Sal and Nic groups (two-way ANOVA, ICSS effect $F_{(1,388)} = 11.53$, $p = 0.0007$), but bursting activity was unchanged after ICSS ($F_{(1,388)} = 0.086$, $p = 0.76$). No interaction effect was observed for firing frequency ($F_{(1,388)} = 1.377$, $p = 0.24$) nor bursting activity ($F_{(1,388)} = 1.691$, $p = 0.19$). **c–f** Comparison between mice exposed to chronic nicotine (Nic, in pink, $n = 27$) and control mice (Ctl, $n = 55$) at the end of the DS, regarding (**c**) the target repartition (i.e., $P_A$, $P_B$, and $P_C$, two-sided Student's $t$-test with Holm correction for multiple comparisons $p = 0.73$, 0.73, and 0.96, respectively), (**d**) the percentage of directional changes (two-sided Student's $t$-test, $p = 0.24$), (**e**) the number of rewards (two-sided Wilcoxon test, $^{***}p = 0.0004$), and (**f**) the cumulative distribution of the average time-to-goal (two-sided KS test, $^{**}p = 0.0003$). **g–j** Comparison between mice exposed to chronic nicotine (Nic, in pink, $n = 27$) and control mice (Ctl, $n = 55$) at the end of the PS, regarding: (**g**) the target repartition. Nic mice visited more often the options with a higher reward probability (i.e., $P_{50}$ and $P_{100}$) and less often the option with the lowest probability ($P_{25}$) in comparison to control mice (two-sided student's $t$-test with Holm correction for multiple comparisons, $^{**}p = 0.006$, $^{*}p = 0.011$, and $^{*}p = 0.012$, respectively). **h** Percentage of directional changes (two-sided Wilcoxon test, $^{*}p = 0.023$); **i** Probability of making the exploitative choice (i.e., the one with the highest probability of reward) for the three possible gambles for nicotine and control mice (two-sided Student's $t$-test with Holm correction for multiple comparisons, $^{*}p = 0.03$) and (**j**) the cumulative distribution of the average time-to-goal (two-sided KS test, $^{**}p = 0.004363$). For (**c–e**) and (**g–i**), data are shown as mean ± sem. Boxplot shown median, quartiles, and extreme values. See also source data.

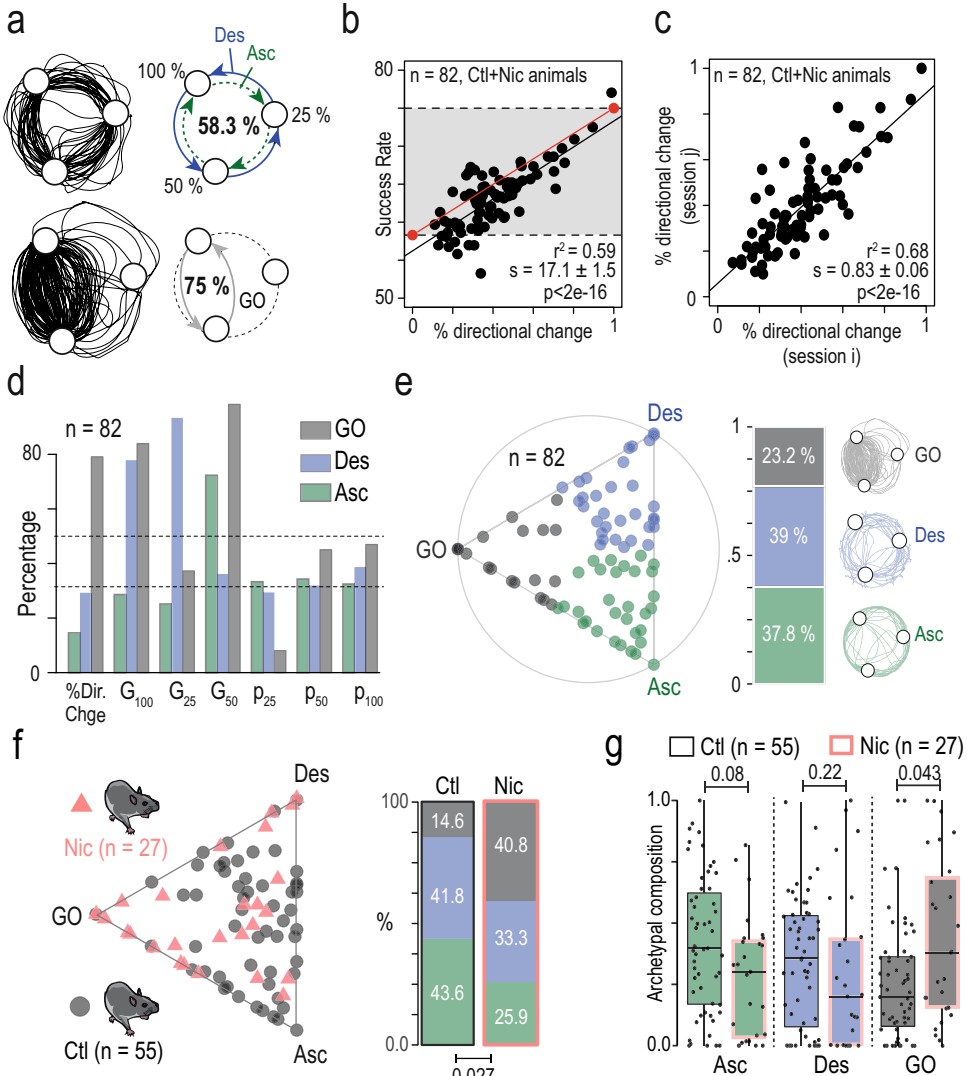

**Fig. 3 Mice exhibited inter-individual differences in choice strategies which were differentially affected by chronic nicotine exposure. a** Left: sample trajectories in the probabilistic setting (PS), corresponding to different choice strategies, a circular strategy (top) and a gain-optimizing strategy (bottom). Right: a mouse using a purely circular strategy (top, descending Des or ascending Asc) in the PS will tend to a 58.3% success rate, whereas a mouse that always avoids $P_{25}$ and alternates between $P_{100}$ and $P_{50}$ (bottom) will reach 75% of success rate. **b** Correlation between the success rate and the percentage of directional changes (linear regression, adjusted $r^2$, the slope estimate $s$ and its $p$ value). Mice displayed a strong inter-individual variability in their choice strategy but, overall, the higher the percentage of directional change, the higher the success rate (regression line in black). The red line indicates the linear correlation passing through two theoretical points: {0% directional changes; 58.3% success rate} and {100% directional changes; 75% success rate}. **c** Correlation between the percentage of directional changes for two consecutive sessions (linear regression, adjusted $r^2$, the slope estimate $s$ and its $p$ value). This measure showed a strong stability between consecutive sessions, indicating that the decision strategy was conserved across time for a given individual. **d, e** Archetypal analysis of the choice strategies based on 7-dimensional data space: (i) the % of directional changes, (ii) the gambles $G_{100}$, $G_{25}$ and $G_{50}$, and (iii) the distribution of choices between $P_{25}$, $P_{50}$, and $P_{100}$. Analysis was performed on $n = 82$ mice (pooled control and nicotine mice). **d** Plot of the three archetypal solutions, gain-maximizers (GO), descending (Des) and ascending (Asc), and their 7 basic variables used in this analysis. **e** Left: visualization of the α coefficients using a ternary plot. Each point represents the projection of an individual onto the plane defined by a triangle where the three apices represent the three archetypes (GO, Des, and Asc). Points are color-coded according to their proximity to the archetypes. Right: proportions of each archetype on the entire population: 37.8% Asc (green), 39% Des (blue), and 23.2% GO (gray). **f** Left: nicotine (Nic, pink triangles) and control (Ctl, gray dots) mice displayed on the same ternary plot. Nic mice displayed a visual shift of their behavior towards the GO extrema of the archetype. Right: this shift was reflected by a difference in the proportion of each phenotype between nicotine and control groups ($\chi^2$ test, $p = 0.027$), with a higher proportion of GO mice in the nicotine group. **g** Archetypal composition for each archetype (1 = closer to the apex) in control and nicotine mice (two-sided Wilcoxon test, $p = 0.08$, $p = 0.22$, and $p = 0.04$, with Holm correction). For (**g**), data are shown as mean ± sem. Boxplot shown median, quartiles, and extreme values. See also source data.

making parameters and session number after the first 5 sessions (Supplementary Fig. 4b). Furthermore, to test whether the variabilities in behavior were robust for each individual from trial to trial, we compared the percentage of directional changes for two consecutive sessions for each animal of the control group.

Directional changes showed a strong positive correlation from one session to the next (Fig. 3c), suggesting a strong consistency in individual behaviors. This observation was generalized by demonstrating that intra-individual variations are lower than the inter-individual variations (Supplementary Fig. 4c).

Having established that inter-individual variations in the PS performance arise from the strategies each mouse adopts within the task, we next aimed to characterize individual behaviors of all mice (both control and nicotine-treated groups, i.e., $n = 82$) in the task. For that purpose, we used a seven-dimensional dataset based on the statistics of (i) the directional changes, (ii) the target distributions and (iii) the three gambles (see data, Fig. 1d–f) followed by archetypal analysis[36,37]. While principal component analysis methods have been classically used to split high-dimensional datasets into clusters by aggregating individual data onto typical observations (the cluster centers), archetypal analysis depicts individual behavior more as a continuum within an "archetypal landscape" defined by extreme strategies: the archetypes. Individual data points are represented as linear combinations of extrema (vertex corresponding to archetypal strategies) of the dataset. The seven-dimensional dataset was used to identify three archetypal phenotypes. The three archetypes and their characteristics (Fig. 3d) differentiated mice exhibiting a GO strategy (i.e., focusing on $P_{50}$ and $P_{100}$) (Fig. 3a, in gray), from mice with circular patterns (equal distribution between the three targets, Fig. 3a), which either turned in a descending manner (labeled Des, in blue, sequence $P_{100}$ - $P_{50}$ - $P_{25}$ associated with high $G_{100}$ and $G_{25}$ but low $G_{50}$) or an ascending manner (labeled Asc, sequence $P_{25}$ - $P_{50}$ - $Pp_{100}$ associated with low $G_{100}$ and $G_{25}$ but high $G_{50}$). The individual behavior of each of the 82 mice could be defined as a weighted combination of these three extrema in a ternary plot (Fig. 3e). An animal's behavior in this ternary plot is defined by three coordinates $(a, b, c)$ that sum to 1 and that depict its relative archetypal composition. Therefore, these coefficients $(a, b, c)$ could be used to assign each individual to its nearest archetype based on its behavioral profile (Fig. 3e, left). This assignment revealed that 23.2% of the mice were closer to the GO archetype (gray), while the remaining mice were evenly distributed between the Des (39%, blue) and Asc archetypes (37.8%, green) (Fig. 3e, right). To analyze the effect of chronic nicotine, we split the control and nicotine-treated mice, and showed that these two groups are distributed differently in the archetypal space as indicated by a modification (i) of the distribution of the archetype's assignments (Fig. 3f) and (ii) of the archetypal composition (Fig. 3g). Overall, chronic nicotine exposure produced an apparent displacement of the population further from Asc and Des apices and closer to the GO apex, thus it favored the emergence of the more exploitative, and thus less explorative, GO phenotype.

**Nicotine modifies decision parameters associated with exploration.** To quantitatively describe the effects of nicotine on the decision processes underlying steady-state choice behavior in mice, we modeled individual data using a softmax model of decision-making. In this model, the probability of choosing target A over B depends on the difference between their expected values, here the probability $P$ of reward delivery associated with each target (as the stimulation magnitudes were the same for all targets), and the "inverse temperature" parameter $\beta$ which represents the sensitivity to the difference of values ($\Delta V$). A small $\beta$ favors exploration (the proportion of respective choices is less sensitive to $\Delta V$, with a null $\beta$ meaning all options have nearly the same probability to be selected, independently of their respective value), while a large $\beta$ indicates exploitation (high sensitivity to $\Delta V$, with an infinite $\beta$ meaning that options associated with higher reward probabilities are always selected). $\beta$ can thus be considered as a proxy to measure the exploration/exploitation trade-off. "Choosing the highest rewarded option" and "exploring less" are therefore equivalent in this exploit/explore framework. This model was adapted to account for the behavior of mice in the PS as follows: first, decisions were biased towards actions with the most uncertain

consequences, by assigning a bonus value $\varphi$ to the expected uncertainties, i.e., the variance $P(1 - P)$ associated with each location[33]. This allowed us to explain the atypically low probability of choosing $P_{100}$ over $P_{50}$ in $G_{25}$ (Fig. 1f). Second, to account for the circular bias observed in both DS and PS, we added a motor cost which decreases the value of a target if it requires the animal to perform a directional change[19]. Thus, in this adapted softmax model (Fig. 4a and "Methods"), three latent variables not directly observed but inferred from the model were used: the "exploration/exploitation" parameter $\beta$, which was defined as the weighted sum of the expected values (100%, 50%, or 25%); expected uncertainty (weighted by parameter $\varphi$); and expected motor cost (weighted by parameter $\kappa$) of a given target.

We fitted the transition function of each mouse from the control group ($n = 55$) with this model, and obtained positive $\beta$, $\varphi$, and $\kappa$ values (Fig. 4b, left). We then compared the output of this model (labeled M3: $\beta > 0$, $\varphi > 0$, and $\kappa > 0$) with two simpler ones, M1 and M2, to test whether mouse choices can be explained by simpler hypotheses. In M1, $\beta$ and $\varphi$ are set to 0, hence choices would be solely driven by motor cost (i.e., a bias against U-turns), which could explain circling behaviors independently of the probabilities associated with reward delivery. In M2 $\varphi$ is set to 0, which would correspond to animals not taking uncertainty into account. Comparison of the models (Bayesian information criterion, Fig. 4b, right; and likelihood ratio test for nested models, Supplementary Fig. 5) indicated that M3 provides the best fit for the data, and suggests that mice used both motor cost, reward probabilities and uncertainty of the reward location to drive their choices.

The generative performance of the model was then assessed by simulating sequences of choices ($n = 2000$ model choices) for $n = 55$ mice with their respective model parameters (Fig. 4c, see also Supplementary Fig. 5). The model accurately reproduced the mean distribution of targets (Fig. 4c, left), the proportion of directional changes (Fig. 4c, middle), and the choice transition function (Fig. 4c, right). Individual transition functions from nicotine-treated mice ($n = 27$) were then fitted by the same model. When compared with the model parameters of control mice, nicotine exposure increased the value sensitivity parameter $\beta$, but did not affect the cost of directional changes ($\kappa$ parameter), nor the uncertainty bonus $\varphi$ (Fig. 4d). We thus asked whether recapitulating these effects on decision parameter $\beta$ would be sufficient to mimic the effect of nicotine. We modeled the choices ($n = 2000$) using decision-making parameters from the control population ($n = 55$, as in Fig. 3b, c) modified by the average difference observed in the $\beta$ parameter from nicotine-treated mice. We compared the three main behavioral measures altered by nicotine: (i) the probability to choose the most valuable option in gamble $G_{100}$ (choosing $P_{50}$ over $P_{25}$), (ii) the percentage of directional changes, and (iii) the probability to visit $P_{25}$. By applying an increase in $\beta$ (derived from nicotine-treated mice) to the control model parameters, the model accurately reproduced the changes observed in decision-making strategy following chronic nicotine exposure for the three measures (Fig. 4e). Conversely, by combining a decrease in $\beta$ (i.e., subtracting the average effect of nicotine from the nicotine-treated model parameters) we were able to simulate the conversion of a nicotine-treated behavioral profile into a control profile. These results thus suggest a specific effect of nicotine on the $\beta$ parameter.

Finally, we assessed the correspondence between the archetypal analysis and the decision-making model, by comparing the value of the three parameters ($\beta$, $\varphi$, $\kappa$) depending on the archetypal composition (see methods). Overall, the three archetypes corresponded to different combinations of the $\beta$ and $\varphi$ model parameters (Fig. 4f), and an almost homogeneous distribution of motor cost $\kappa$. The GO (gray) archetype was associated with a high

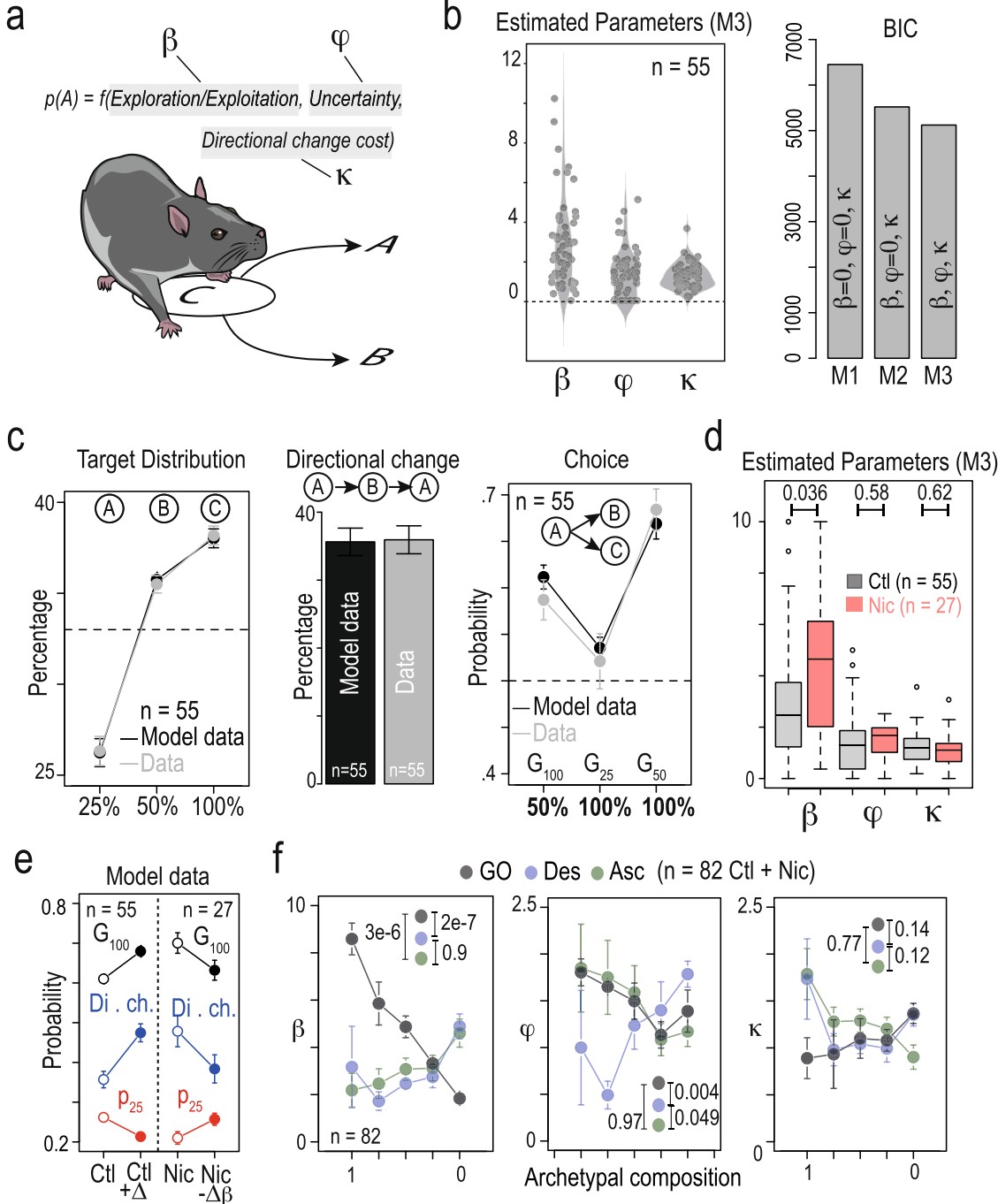

value of β (corresponding to exploitation) and φ, which is consistent with individuals that favor the alternation between locations associated with higher probability ($P_{100}$ and $P_{50}$). The Des and Asc phenotypes corresponded to strong circular behaviors and a low sensitivity to value (low β value), resulting in an important impact of the motor cost parameter (κ) in their strategies. The Des and Asc groups differed by their preference to uncertainty, φ, value (Δ = 1.012, $p = 0.0079$), which was related to the directionality of their preferred rotation: a low φ corresponds to mice choosing the certain ($P_{100}$) reward over the uncertain ($P_{50}$) reward, resulting in a tendency for sequence $P_{25}$ -> $P_{100}$ -> $P_{50}$ observed in Des mice (blue). Conversely, a high φ is associated with the reverse sequence $P_{25}$ -> $P_{50}$ -> $P_{100}$ observed in Asc mice (green). Such decomposition of the archetypal phenotypes into their underlying decision-making

processes illustrates how distribution of individual decision-making strategies (Asc, Des, and GO) in a population could correspond to transitions in the parameter values from the same model. Overall, the model identifies the effect of nicotine as an increase in β, which is consistent with a displacement of exposed mice towards the GO profile in the archetypal space.

**Optogenetic DA neuron stimulation recapitulates the effects of nicotine.** Nicotine exposure is known to induce modifications in a number of brain areas[38], including an increase in the tonic activity of VTA DA neurons, as we indeed confirmed in this study (Fig. 2b). Furthermore, the tonic activity of DA neurons has been proposed to play a role in the balance between exploration and exploitation[29–31]. We thus asked whether directly and acutely

**Fig. 4 Computational modeling suggests that decision parameters differ between the three archetypes and are differentially affected by nicotine exposure. a** Principle of the softmax model: softmax decision rule with three parameters β (inverse temperature or exploration/exploitation), φ (uncertainty bonus), and κ (cost or effort for a directional change). **b** Left: estimated values of β, φ, and κ parameters (individual points and density) with M3 model for the 55 control mice (not exposed to nicotine). Right: Bayesian information criterion (BIC) computed using three models of choice selection (Softmax with β = 0, φ = 0, κ > 0 (M1), β > 0, φ = 0, κ > 0 (M2), and β > 0, φ > 0, κ > 0 (M3), see "Methods"). Smaller BIC value indicates that the uncertainty bonus provided a better fit. **c** Comparison between control data and model for a model sequence of 2000 choices (n = 55) simulated with fitted values of β, φ, and κ (see **b**) Left: repartition of visits on the three targets ($P_{25}$, $P_{50}$, and $P_{100}$, with a mean of the differences between control data and model of Δ = 0.002%, −0.003% and 0.001%, two-sided Student's t-test with Holm correction for multiple comparisons p = 1). Middle: Comparison of the percentage of directional changes (Δ = 0.002, p = 0.93). Right: probability to choose alternatives with the highest probability of reward for the three possible gambles ($G_{100} = P_{50}$ over $P_{25}$; $G_{25} = P_{100}$ over $P_{50}$; $G_{50} = P_{100}$ over $P_{25}$, Δ = −0.02, −0.008, and 0.009, two-sided Student's t-test with Holm correction for multiple comparisons p = 1, for the three gambles). **d** Nicotine-exposed animals displayed an increase in β (Δ = 1.73, p = 0.036), but no difference in φ (Δ = 0.09, p = 0.58) and in κ (Δ = −0.07, p = 0.62) compared to control mice (two-sided Wilcoxon test with Holm correction for multiple comparisons). **e** Mimicking the effect of nicotine on the model parameters. Left: the simulation of choice behavior when nicotine-induced increase of β is added to the control model parameters (n = 55, Ctl + Δβ) recapitulates the effect of nicotine on the three choice parameters (the probability to choose the most valuable option in gamble $G_{100}$; the percentage of directional changes, and the probability to visit $P_{25}$, mean of the differences between nicotine (Nic) data and model Δ = 0.007%, 0.0021%, −0.0008%, respectively, two-sided Student's t-test with Holm correction for multiple comparisons, p = 1). Starting from the nicotine mice parameters and removing the nicotine-induced changes on β (n = 27, Nic −Δβ) reestablish those three parameters at the level of control mice (mean of the differences between control data and model Δ = −0.044%, −0.024%, 0.006%, respectively, two-sided Student's t-test with Holm correction for multiple comparisons, p = 0.62, 1, and 1, respectively). Δβ is calculated using $β_{Nic}$ – $β_{Ctl}$ the mean estimated in control and nicotine condition. **f** Left: correlation between β (left), φ (middle), or κ (right) values and the archetypal composition for both control and nicotine mice (n = 82, see plot in Fig. 3b). The closer to the GO phenotype, the higher the β, which is consistent with an optimal strategy based on alternation between $P_{100}$ and $P_{50}$. The closer to the Des phenotype, the lower the φ parameter. Statistics (inset) compares β, φ, and κ in three groups obtained from the entire populations (n = 82) classified depending on the archetype proximity (two-sided Wilcoxon test with Holm correction for multiple comparisons). For (**c**–**f**), data are shown as mean ± sem. See also source data.

modifying the activity of VTA DA neurons is sufficient to alter decision-making behavior within a session and recapitulate the effects of chronic nicotine in our ICSS task. To specifically and bi-directionally manipulate VTA DA neurons, we expressed either an excitatory channelrhodopsin (CatCh[39]) or an inhibitory halorhodopsin variant (Jaws[40]) in DAT[iCRE] mice using a Cre-dependent viral strategy (Supplementary Fig. 5a). We confirmed in patch-clamp recordings that continuous 5 ms-light pulses at 8 Hz (470 nm) reliably increased VTA DA neuron activity in CatCh-transduced mice (Fig. 5a), while 500 ms-light pulses at 0.5 Hz (520 nm) reliably decreased their activity in Jaws-transduced mice (Fig. 5b). We then specifically tested the hypotheses that optogenetic activation of VTA DA neurons should reproduce the increased exploitation seen in nicotine-treated animals, and that optogenetic inhibition should produce the opposite effect.

After mice completed both the DS and PS in the ICSS task, they went through optogenetic sessions maintaining the PS rules, with an alternating schedule of 2 days with photo-stimulation (ON, photo-stimulation started 5 min prior to the start of the daily session and was maintained throughout the 5 min session) and without (OFF) (Fig. 5c). During the OFF days, mice were connected to the optical fiber patch-cord but did not receive light stimulation. For each pair of ON/OFF experiments, we estimated the effect of the photo-stimulation on the four main behavioral measures that were altered by chronic nicotine (Fig. 5d and Supplementary Fig. 6d). As expected, optogenetic activation increased directional changes (Fig. 5d, left) and decreased the probability to visit $P_{25}$ (Fig. 5d, right), favoring alternations between $P_{100}$ and $P_{50}$, similar to the effect of nicotine. Opposite effects were observed for these two parameters when the firing rate was reduced in VTA DA cells using Jaws (Fig. 5d). Optogenetic activation reduced time-to-goal without affecting the choice in the gamble $G_{100}$, while optogenetic inhibition did not significantly affect either of these two parameters (Supplementary Fig. 6d). We fitted the transition function of CatCh- and Jaws-transduced mice with our decision-making model. The effects of photo-activating VTA DA neurons on decision-making during the ICSS task could be modeled as an increase of β

(Fig. 5e), as observed with chronic nicotine. Photo-inhibition of VTA DA neurons, however, did not significantly affect the exploration/exploitation trade-off parameter β (Fig. 5e). For each pair of ON/OFF experiments, we also estimated the effect of the photo-stimulation on our seven main behavioral measures (Fig. 1d–f) and β parameter by calculating for a given measure (M) the difference $Δ = M_{ON} − M_{OFF}$. These differences were compared with the net effect of nicotine obtained for each of these parameters by subtracting the mean control value from the mean effect of nicotine (Supplementary Fig. 6e, red). Overall, nicotine and photo-stimulation produced a similar pattern of effects on our behavioral measure (Supplementary Fig. 6e), while inhibition produced the opposite effect albeit to a lesser extent.

Finally, by analyzing decision-making behaviors between the stimulated (ON) and non-stimulated (OFF) conditions in the previously identified archetypal space, we revealed that VTA DA neuron activation draws individual phenotypes towards the GO archetype (i.e., increased GO archetypal composition), while VTA DA neuron inhibition draws individuals away from GO (Fig. 5f). Thus, altering the firing pattern of VTA DA neurons is, by changing both the motor cost and the balance between exploration and exploitation behavior, sufficient to bias decision behaviors in the ICSS task, as suggested by our simulations (Fig. 4). Furthermore, increasing VTA DA neuron firing mimicked the effects of chronic nicotine exposure on decision-making measures, linking the behavioral alterations with the physiological changes to DA neurons we observed in nicotine-treated mice.

## Discussion

Understanding how nicotine affects decision-making has been challenging, because two different physiological aspects need to be distinguished[41]: (i) nicotine as a reinforcer that directly activates the dopaminergic system to produce reinforcement and nicotine-seeking, and (ii) nicotine as a neuromodulator that alters nicotine-independent decision-making processes by modifying the dynamics and computational properties of cholinoceptive circuits. Here, using a multi-armed ICSS bandit task, we show that mice passively treated with nicotine forage more frequently

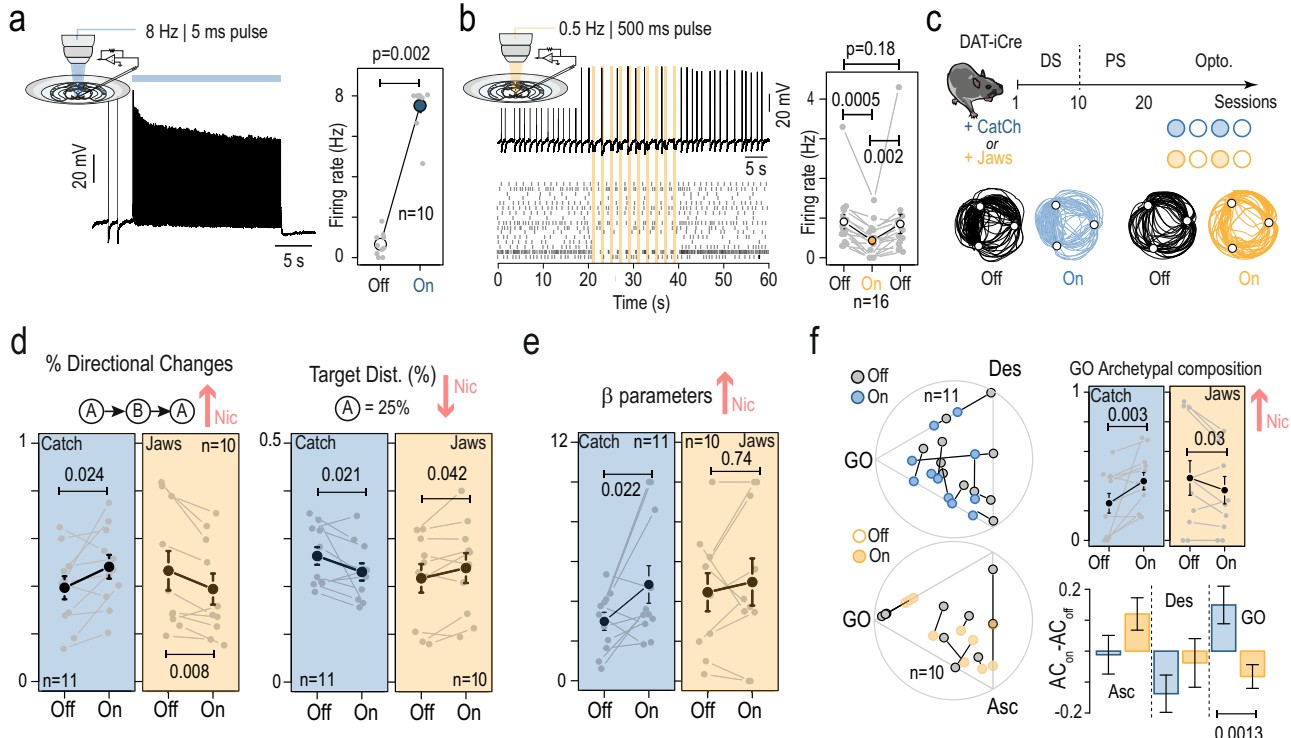

**Fig. 5 Optogenetic manipulation of VTA DA neuron activity recapitulated the behavioral adaptations observed under chronic nicotine exposure. a** Left: representative ex vivo current-clamp recording of a VTA DA neuron transduced with CatCh and stimulated with 5 ms blue light pulses at 8 Hz. Right: average increase in basal firing frequency upon optogenetic stimulation for $n = 10$ neurons ($p$ value $= 0.002$, two-sided Wilcoxon test). **b** Top left: representative ex vivo current-clamp recording of a VTA DA neuron transduced with Jaws and stimulated with 500 ms green light pulses at 0.5 Hz. Bottom left: raster plot for $n = 16$ neurons. Right: average decrease in basal firing frequency upon optogenetic stimulation, and return to the baseline after the photo-stimulation period, for $n = 16$ neurons ($p$ value: ns $= 0.18$; **0.004; **0.0014, two-sided Wilcoxon test with Holm correction). **c** Task design and photo-stimulation protocols. DAT$^{iCRE}$ mice transduced with either an AAV-DIO-CatCh-YFP in the VTA (CatCh, blue) or an AAV-DIO-Jaws-eGFP (Jaws, yellow) and were implanted unilaterally with bipolar stimulating electrodes for ICSS in the MFB (see also Supplementary Table 1). Following the deterministic (DS) and probabilistic setting (PS) sessions, they received 2 paired Off (filled circles) and Off (open circles) sessions with the same rules as in the PS. Below: representative trajectories of a CatCh and Jaws-transduced mouse with (blue and yellow) and without (black) optogenetic stimulation of VTA DA neurons. **d** Left: percentage of directional changes and (right) target distribution (P$_{25}$) in individual mice (gray points) for CatCh ($n = 11$) and Jaws-transduced ($n = 10$) mice during On and Off sessions. Mean + sem are in black. Red arrows indicate the net effect of nicotine for comparison (one-sided Student's $t$-test or Wilcoxon test). **e** Same as (**d**) for the softmax model $\beta$ parameter. **f** Left: position of each animal in the ternary archetype plot for CatCh ($n = 11$, above) and Jaws-transduced ($n = 10$, below) mice, in On (blue or yellow points) and Off (gray point) sessions. Right: (above) GO archetypal composition in individual mice (gray points) during On and Off session, for CatCh ($n = 11$) and Jaws-transduced ($n = 10$) mouse. Mean + sem are in black; (below) net effect (ON–OFF) of light stimulation in archetypal composition for each archetype. Optogenetic activation of DA neurons triggered a shift of the behavior towards the GO phenotype while optogenetic inhibition induced a shift of the behavior away from GO (one-sided Student's $t$-test). For (**a**, **b**) and (**d**, **f**), data are shown as mean ± sem. See also source data.

at locations with the highest probabilities of reward (P$_{50}$ and P$_{100}$) compared to non-exposed animals, suggesting a bias in the exploration/exploitation trade-off toward decreased exploration. These behavioral changes were accompanied by modifications in the spontaneous activity of VTA DA neurons. We further showed that inter-individual variations in foraging strategies emerge between mice, despite the fact that they are all males of the same genetic background. This suggests that animals idiosyncratically adapt their behavior in response to task constraints, rather than all converging toward a theoretical "optimal" performance maximizing reward. Finally, optogenetically increasing or decreasing VTA DA neuron activity shifted individual strategies, recapitulating the results from nicotine-exposed mice and computational modeling. Together, our data suggest that modifications of the dopaminergic activity, notably through chronic nicotine exposure, scale the exploration/exploitation trade-off.

In our task, mice adapted their choices according to the probability of reward delivery, but they also consistently kept visiting the targets associated with lower reward probabilities in

all of the gambles, even after extended training. Such a high level of exploratory behavior is potentially attributable to the setup, which is characterized by the delivery of small rewards, serially repeated gambles with short delays between trials, and learning through experience[42]. In the exploration/exploitation framework, the fact that mice continue to visit targets with the lowest reward probability in each of the gambles, despite intensive learning, can reflect (i) exploratory noise, generally modeled via decreased value sensitivity (or increased randomness) $\beta$ in the softmax model, (ii) directed exploration, if one considers that mice continue to explore locations associated with low reward probability to reduce the uncertainty associated with probabilistic omission and gain information of task contingencies, and (iii) uncertainty-seeking, which is neither really explorative nor exploitive, but considers that mice simply attribute a positive value to expected uncertainty, like a bonus for playing or gambling. Our analysis also introduces the idea of qualitative inter-individual variations, sometimes called "idiosyncrasy", in choice strategy. Model comparison first suggests that all mice, even those that are away from

the GO archetype, used information about reward probabilities and uncertainty. It also shows that the inter-individual variations were well described by a single computational model of decision-making that takes into account the exploration/exploitation trade-off, uncertainty, motor cost, and continuous variations of two latent variables[43] inferred through the model. Note that sex or strain differences may constitute another layer of variability[44], which we are currently addressing in ongoing experiments. Finally, despite variations in individual choice behaviors, the consequences of nicotine administration were consistent, with a clear effect on the β parameter, and a strategy biased towards the exploitation of the highest reward values.

This interpretation is supported by two findings. Firstly, we could demonstrate that variations in behavior are not due to chance but indicate individual personalities as revealed by the strong correlation in descriptive parameter values between consecutive sessions (Fig. 4b). Secondly, it is conceivable that mice eventually achieve the optimal behavior (alternating solely between $P_{50}$ and $P_{100}$), albeit at different rates of learning. In this case, a possible interpretation of our data would be that nicotine facilitates learning[45] and speeds up the convergence toward the GO profile. However, this hypothesis is unlikely, and is not supported by our data. Indeed, despite significant adaptations during the transition between DS and PS, as well as during the first deterministic and probabilistic sessions (1–4), the animals' behavior at the end of each setting is overall stable, and the choice behavior close to steady-state. In addition, manipulating the activity of VTA DA neurons using optogenetics acutely altered the behavioral strategies of the mice, with kinetics that are incompatible with synaptic plasticity or learning. We thus argue that inter-individual variations in task performance, as well as nicotine effects, should not be interpreted as differences in learning processes or knowledge of the environment, but rather as differences in using the knowledge acquired about the statistical structure of the environment (quantified by variations in β, φ, and κ) to develop their strategy within the task.

The increase of β reflects an amplified exploitative behavior, an effect that has been previously linked to enhanced tonic DA activity, which is hypothesized to modulate the bias towards optimal choices[29–31]. In this study, we demonstrate a direct link between the tonic activity of DA neurons and exploitation using electrophysiological and optogenetic approaches. The multi-armed ICSS bandit task enables, through a clear distinction between action selection (choices) and action execution (time-to-goal), to identify the modified components of value-based decision-making in relation to tonic DA. We explicitly demonstrate an increase in value sensitivity due to nicotine-induced alterations in tonic DA activity. Previous ICSS studies have observed that chronic exposure to drugs sensitizes the brain reward system, and in doing so lowers the stimulation threshold (expressed as a current intensity or stimulation frequency)[46] required for ICSS[47]. Here we expand these results by quantifying the effects of such increased value sensitivity on choices between ICSS-mediated rewarding locations, and further identifying a causal link between these behavioral modifications and increased tonic activity of VTA DA neurons.

In the context of DA neuron physiology, activity varies in frequency and in degree of burst firing. Bursting has been defined as successive action potentials separated by less than 80 ms[48], occurring on top of a regular "pacemaking" firing activity. Spontaneous activity (regular spiking and bursting) is associated with the neuromodulatory function of DA and its ability to shift ongoing dynamics of target structures. In this context, bursting is not necessarily locked to any behaviorally relevant or salient event. By contrast, phasic activity is related to event-locked increase in firing[22], which can typically be observed as a synchronous increase in firing rate in a population of

neurons, but is not necessarily composed of bursts of action potentials (i.e., it can be single spikes but time-locked to an event during successive trials). DA phasic activity modulates synaptic plasticity[49] and is critical for learning the value of stimulus or actions[50]. The observed increase in VTA DA neuron activity (both in bursts and in firing rate) after nicotine exposure suggests that dopaminergic tone is modified in nicotine-exposed animals. Such an increase in the basal activity of VTA DA neurons[9,35] occurs through desensitization and up-regulation of nicotinic receptors and the long-term strengthening of glutamatergic synaptic transmission[51]. Here we show that acutely elevating VTA DA neurons activity using optogenetics is sufficient to mimic the behavioral alterations seen in mice under chronic nicotine exposure. Nicotine and optogenetic stimulation obviously act differently on DA neurons, yet they both lead to an increase in VTA DA neurons activity. Our optogenetic experiments confirmed once more that acutely modifying DA neurons activity did not change the animal's knowledge of reward probabilities (learning), but the way the animals used the learned contingencies (values, probabilities and uncertainties) to shape a decision strategy or policy. We thus link DA tonic neuromodulatory function and modifications of decision-making parameters (here β).

Variations in neuromodulatory functions, including those in the catecholamine and cholinergic systems, contribute to the process of individuation[52–54]. DA, and in particular from the VTA, has been linked to a cluster of traits (extraversion, novelty-seeking, etc.) conceptually related to reward-seeking[55,56]. However, despite the substantial attention paid to DA in personality neuroscience, and despite a clear association between modulations in dopaminergic function and variations in individual traits, defining which specific traits are influenced by DA remains a challenging task. Our data suggest that modification in basal VTA DA neuron activity can directly modify the expression of a specific trait: the exploit/explore trade-off here estimated through the β variable. This result is reminiscent of the observations made from male mice living continuously in a large environment, which display idiosyncratic behavioral strategies during a decision-making task, and for which the exploration/exploitation balance was correlated with the activity of their DA system[54].

Nicotine exposure alters decision-making processes[6]. Non-contingency studies have previously shown that yoked nicotine exposure increases the incentive salience of non-nicotine stimuli[57], similar to the sensitization to ICSS rewards[47]. These studies suggest an essential role of contextual cues in smoking and the nicotine-induced increase in reward sensitivity. Neuroeconomics studies have also linked smoking with increased impulsivity (delay-discounting task[11]), lack of counterfactual learning signals[58], and decreased behavioral flexibility (exploration in a dynamic bandit task[13]). Our results further reveal that nicotine exposure decreases exploration. In addition, we provide a mechanistic understanding of how reward processing may be altered at the level of the VTA in response to chronic nicotine. This insight is translationally valuable as nicotine-induced alterations in explore/exploit processes likely also have implications for the everyday life of smokers, particularly as they can increase vulnerability for addiction to other drugs of abuse and for behavioral disorders such as pathological gambling that rely on value-based decisions[7,59] and present a high comorbidity with tobacco addiction[60]. Our data underscores altered choice behaviors in smokers that likely participate in, but are not limited to, addiction[6]. Finally, such an explore–exploit paradigm and archetypal analysis could be very useful to study the effects of other drugs of abuse on decision-making. Indeed, humans with methamphetamine[61] or alcohol use disorders[62] display alterations in bandit tasks, but human studies cannot disambiguate whether altered decision-making facilitates, or results from, drug use. Hence, preclinical studies are needed to dissect the causal

mechanisms underlying alterations in economic decisions, and to understand the dynamics of drug users' profiles in general, and of smokers (or vape users) in particular.

## Methods

**Animals.** Experiments were performed on adult C57Bl/6Rj DAT[iCRE] and wild-type (Janvier Labs, France) mice. Male mice, from 8 to 16 weeks old, weighing 25–35 g, were used for all the experiments. They were kept in an animal facility where temperature ($20 \pm 2$ °C) and humidity were automatically monitored and a circadian light cycle of 12/12 h light–dark cycle was maintained. All experiments were performed in accordance with the recommendations for animal experiments issued by the European Commission directives 219/1990, 220/1990, and 2010/63, and approved by Sorbonne University.

**AAV production.** AAV vectors were produced as previously described[63] using the cotransfection method and purified by iodixanol gradient ultracentrifugation[64]. AAV vector stocks were tittered by quantitative PCR (qPCR)[65] using SYBR Green (Thermo Fischer Scientific). Additional information is provided in Supplementary Table 1.

**Intracranial self-stimulation (ICSS) electrode implantation.** Mice were anaesthetized with a gas mixture of oxygen (1 L/min) and 1–3% of isoflurane (Piramal Healthcare, UK), then placed into a stereotaxic frame (Kopf Instruments, CA, USA). After the administration of a local anesthetic (Lurocain, 0.1 mL at 0.67 mg/kg), a median incision revealed the skull which was drilled at the level of the median forebrain bundle (MFB). A bipolar stimulating electrode (PlasticOne 2-channels, stainless-steel, 10 mm) for ICSS was then implanted unilaterally (left or right, randomized) in the brain (stereotaxic coordinates from bregma according to mouse after Paxinos atlas: AP −1.4 mm, ML ±1.2 mm, DV −4.8 mm from the brain). Dental cement (SuperBond, Sun Medical) was used to fix the implant to the skull. After stitching and administration of a dermal antiseptic, mice were then placed back in their home-cage and had, at least, 5 days to recover from surgery. An analgesic, buprenorphine solution at 0.015 mg/L (0.1 mL/10 g), was delivered after the surgery and if necessary, the following recovering days. The efficacy of electrical stimulation was verified through the rate of acquisition during the DS (see Intracranial Self Stimulation (ICSS) bandit task).

**Implantation of osmotic minipumps.** After 5 days of training in the DS (see Behavioral methods), animals were anesthetized with a gas mixture of oxygen (1 L/min) and 1–3% of isoflurane (IsoVet, Piramal Healthcare, UK). After the administration of a local anesthetic, an incision was performed at the level of the interscapular zone, to subcutaneously implant an osmotic minipump (Model 2004, ALZET, CA, USA) containing 200 μL of either a solution of nicotine hydrogen tartrate salt (Sigma-Aldrich, USA) at a dose of 10 mg/kg/day (free base) or saline solution ($H_2O$ with 0.9% NaCl) for the control group. Both solutions were prepared in the laboratory. Minipumps delivered their content with a flow rate of 0.25 μL/h over 28 days (covering the remaining training days in the DS and all sessions in the PS). The surgical wound was closed with surgical stitches. Animals had 2 days of rest to recover from the minipump surgery before going further with their behavioral training.

**Virus injections and optogenetics experiments.** DAT[iCRE] mice were anaesthetized (Oxygen 1 L/min, Isoflurane 1–3%) and implanted with an ICSS electrode as described above. They were then injected unilaterally (randomized left/right side and ipsi/contralateral side regarding the ICSS electrode) in the VTA (1 μL, coordinates from bregma: AP −3.1 mm; ML ±0.5 mm; DV −4.55 mm from the skull) with an adeno-associated virus (see Supplementary Table 1; AAV5.Ef1α.-DIO.hCatCh.YFP, AAV5.Ef1α.DIO.Jaws.eGFP or AAV5.Ef1α.DIO.YFP). A double-floxed inverse open reading frame (DIO) allowed to restrain the expression of CatCh ($Ca^{2+}$-translocating channelrhodopsin) or Jaws (red-shifted cruxhalorhodopsin) to VTA dopaminergic neurons of DAT[iCRE] mice.

For optogenetic experiments on freely moving mice, an optical fiber (200 μm core, NA = 0.39, Thor Labs) coupled to a ferule (1.25 mm) was implanted just above the VTA ipsilateral to the viral injection (coordinates from bregma: AP −3.1 mm, ML ±0.5 mm, DV 4.4 mm from the skull), and fixed to the skull with dental cement (SuperBond, Sun Medical). The behavioral task began at least 4 weeks after virus injection to allow the transgene to be expressed in the target DA cells. An ultra-high-power LED (470 nm or 520 nm, Prizmatix) coupled to a patch cord (500 μm core, NA = 0.5, Prizmatix) was used for optical stimulation (output intensity of 10 mW). Optical stimulation was delivered continuously, starting 5 min before and continuing throughout the 5 min of ON sessions of the task. Excitatory opsin (CatCh) was stimulated using 470 nm light pulses of 5 ms duration and 8 Hz frequency. Inhibitory opsin (Jaws) was stimulated using 520 nm light pulses of 500 ms duration and 0.5 Hz frequency. The experiment followed a schedule of paired ON and OFF days after the end of training phase (DS + PS). The optical stimulation patch cord was plugged onto the ferrule during all experimental sessions (ON and OFF days) to habituate animals and control for latent experimental effects.

**Ex vivo patch-clamp recordings of VTA DA neurons.** To verify the functional expression of the excitatory opsin CatCh and the inhibitory opsin Jaws, 10–12 weeks-old male DAT[iCRE] mice were injected with the viruses described above. After 4 weeks, mice were deeply anesthetized with an intraperitoneal (IP) injection of a mix of ketamine/xylazine. Coronal midbrain sections (250 μm) were sliced using a Compresstome (VF-200; Precisionary Instruments) after intracardiac perfusion of cold (4 °C) sucrose-based artificial cerebrospinal fluid (SB-aCSF) containing (in mM): 125 NaCl, 2.5 KCl, 1.25 $NaH_2PO_4$, 5.9 $MgCl_2$, 26 $NaHCO_3$, 25 Sucrose, 2.5 Glucose, and 1 Kynurenate (pH 7.2, 325 mOsm). After 10–60 min at 35 °C for recovery, slices were transferred into oxygenated aCSF containing (in mM): 125 NaCl, 2.5 KCl, 1.25 $NaH_2PO_4$, 2 $CaCl_2$, 1 $MgCl_2$, 26 $NaHCO_3$, 15 Sucrose, and 10 Glucose (pH 7.2, 325 mOsm) at room temperature for the rest of the day and individually transferred to a recording chamber continuously perfused at 2 mL/min with oxygenated aCSF. Patch pipettes (4–8 MΩ) were pulled from thin wall borosilicate glass (G150TF-3, Warner Instruments) using a micropipette puller (P-87, Sutter Instruments, Novato, CA) and filled with a KGlu-based intra-pipette solution containing (in mM): 116 K-gluconate, 10-20 HEPES, 0.5 EGTA, 6 KCl, 2 NaCl, 4 ATP, 0.3 GTP, and 2 mg/mL biocytin (pH adjusted to 7.2). Transfected VTA DA cells were visualized using an upright microscope coupled with a Dodt contrast lens and illuminated with a white light source (Scientifica). To characterize CatCh expression, a 460 nm LED (CoolLED) was used both for visualizing YFP-positive cells (using a band-pass filter cube) and for optical stimulation through the microscope (1 s continuous for light-evoked current in voltage-clamp mode and 8 Hz with 5 ms/pulse to drive neuronal firing in current-clamp mode). Regarding Jaws expression, 20 s continuous photo-stimulation of 500 ms pulses at 0.5 Hz with a 525 nm, pE-2, CoolLED, was used in current-clamp (−60 mV). Whole-cell recordings were performed using a patch-clamp amplifier (Axoclamp 200B, Molecular Devices) connected to a Digidata (1550 LowNoise acquisition system, Molecular Devices). Signals were low-pass filtered (Bessel, 2 kHz) and collected at 10 kHz using the data acquisition software pClamp 10.5 (Molecular Devices). All the electrophysiological recordings were extracted using Clampfit (Molecular Devices) and analyzed with R.

**In vivo juxtacellular recordings of VTA DA neurons.** Mice were deeply anaesthetized with chloral hydrate (8%), 400 mg/kg IP, supplemented as required to maintain optimal anesthesia throughout the experiment. The scalp was opened and a hole was drilled in the skull above the location of the VTA. Extracellular recording electrodes were constructed from 1.5 mm O.D./1.17 mm I.D. borosilicate glass tubings (Harvard Apparatus) using a vertical electrode puller (Narishige). Under microscopic control, the tip was broken to obtain a diameter of approximately 1 μm. The electrodes were filled with a 0.5% NaCl solution containing 1.5% of Neurobiotin tracer (AbCys) yielding impedances of 6–9 MΩ. Electrical signals were amplified by a high-impedance amplifier (Axon Instruments) and monitored through an audio monitor (A.M. Systems Inc.). The signal was digitized, sampled at 25 kHz and recorded using Spike2 software (Cambridge Electronic Design) for later analysis. The electrophysiological activity was sampled in the central region of the VTA (coordinates from bregma: 3.1–4 mm AP, 0.3–0.7 mm ML, and 4–4.8 mm DV from the brain surface). Individual electrode tracks were separated from one another by at least 0.1 mm in the horizontal plane. Spontaneously active DA neurons were identified based on previously established electrophysiological criteria[66,67].

**Fluorescence immunohistochemistry.** After euthanasia, induced by IP injection of euthasol (0.1 mL per 30 g at 150 mg/kg) or by paraformaldehyde (PFA) intracardiac perfusion, brains were rapidly removed and fixed in 4% PFA. Following a period of fixation at 4 °C, serial 60 μm sections were cut from the midbrain with a vibratome. Immunohistochemistry was performed as follows: free-floating VTA brain sections were incubated 1 h at 4 °C in a blocking solution of phosphate-buffered saline (PBS) containing 3% bovine serum albumin (BSA, Sigma A4503) and 0.2% Triton X-100 and then incubated overnight at 4 °C with a mouse anti-tyrosine hydroxylase antibody (TH, Sigma, T1299) at 1:500 dilution in PBS containing 1.5% BSA and 0.2% Triton X-100 (see supplementary Table 1). The following day, sections were rinsed with PBS and then incubated for 3 h at 22–25 °C with Cy3-conjugated anti-mouse (Jackson ImmunoResearch, 715-165-150) at 1:500 dilution in a solution of 1.5% BSA in PBS, respectively. After three rinses in PBS, slices were wet-mounted using Prolong Gold Antifade Reagent (Invitrogen, P36930). Microscopy was carried out with a fluorescent microscope Leica DMR, and images captured in gray level using MetaView software (Universal Imaging Corporation) and colored post-acquisition with ImageJ.

For the optogenetic experiments on DAT[iCRE] mice, an immunohistochemical identification of the transfected neurons was performed as described above, with an addition of chicken anti-eYFP antibodies (Life technologies Molecular Probes, A-6455) at 1:1000 dilution (Supplementary Table 1). A goat anti-chicken AlexaFluor 488 secondary antibody (711-225-152, Jackson ImmunoResearch) at 1:1000 dilution was then used in a solution of 1.5% BSA in PBS. Neurons co-labeled for TH and YFP in the VTA allowed to confirm their neurochemical phenotype and the transfection success.

## Intracranial self-stimulation (ICSS) bandit task

*Behavioral setup.* The ICSS bandit task took place in a circular open-field with a diameter of 68 cm. Three explicit square-shaped marks ($1 \times 1$ cm) were placed in the open field, forming an equilateral triangle (side = 35 cm). Entry in the circular zones (diameter = 6 cm) around each mark was associated with the delivery of a rewarding ICSS stimulation. Experiments were performed using a video camera, connected to a video-tracking system, out of sight of the experimenter. A LabVIEW (National Instruments) application precisely tracked and recorded the animal's position with a camera (20 frames/s). When a mouse was detected in one of the circular rewarding zones, an electrical stimulator received a TTL signal from the software application and generated a 200 ms train of 5 ms biphasic square waves pulsed at 100 Hz (20 pulses per train). ICSS intensity was adjusted, within a range of 20–200 µA, during training (see "Training settings") and then kept constant, so that mice would achieve between 50 and 150 visits per session (5 min duration) for two successive sessions, and then kept constant for all the experiment. Mice with insufficient scores in the DS and PS (<40 visits despite increasing the maximum intensity to 200 µA) were excluded.

*Training settings.* The training consisted of two settings: the deterministic setting (DS) and the probabilistic setting (PS), both consisting of at least 10 daily sessions of 5 min. In the DS, all zones were associated with an ICSS delivery ($P = 100\%$). However, two consecutive rewards could not be delivered on the same target, which motivates mice to alternate between targets. In the PS, the zones were associated with three different probabilities ($P = 25\%$, $P = 50\%$, $P = 100\%$) to obtain an ICSS stimulation. The probabilities locations were pseudo-randomly assigned per mouse.

*Data acquisition per experimental group.* Different experimental groups underwent the ICSS bandit task. Firstly, locomotion and choice behavior of the mice, which had been implanted with osmotic minipumps (saline = 23, nicotine = 27), were analyzed and compared between the last 2 days of both training settings. For optogenetics experiments, the DAT[iCRE] mice ($n = 21$) completed the training, followed by a schedule of paired sessions with photo-stimulation (ON) alternated with days without photo-stimulation (OFF). The control animals ($n = 55$) were obtained by pooling together mice implanted with a saline minipump ($n = 23$) and non-implanted mice ($n = 32$). Figure 1 used only data from the non-implanted mice group. Figs. 2, 3, and 4 used the pooled control group.

*Behavioral measures.* For all of those groups, the following measures were analyzed and compared in the PS, as well as in the DS for the saline vs nicotine experiment: (i) number of visits, (ii) time-to-goal, (iii) choice repartition (proportion of visits at each location $P_{25}$, $P_{50}$, and $P_{100}$), and (iv) percentage of directional changes (*n*th visit = *n*th visit + 2). Furthermore, the ICSS bandit task can be seen as a Markovian decision process. Every transition between zones can be considered as a binary choice between two options, since the occupied zone cannot be reinforced twice in a row. The sequence of choices per session is summarized by the proportional result of the sum of three specific binary choices (or gambles, e.g., $G_C$ would be the total number of visits in target A/total number of visits in targets A and B, when the animal is in target C). The three gambles (G) were named after the point on which the mouse is positioned at the time of this gamble: $G_{25} = 100\%$ vs $50\%$, $G_{100} = 50\%$ vs $25\%$ and $G_{50} = 100\%$ vs $25\%$. The target selected in these gambles reflects the balance between exploitative (choosing the most valuable option) and exploratory (choosing the least valuable option) choices. With a softmax-based decision-making model fitted in the laboratory, we computed three parameters: the value sensitivity or inverse temperature (the power to discriminate between values in a binary choice), the uncertainty bonus (the preference for expected uncertainty, considering the reward variance of every option in a binary choice) and the motor cost to do a directional change (target value decreases if it requires to go back to the previous target).

*Decision model.* Decision-making models determined the probability $P_i$ of choosing the next state $i$, as a function (the "choice rule") of a decision variable. Because mice could not return to the same rewarding target, they had to choose between the two remaining ones. Accordingly, we modeled decisions between two alternatives labeled A and B and used a "softmax" choice rule, defined by $P_A = 1/(1 + e^{-ß(vA-vB)})$, where $ß$ is an inverse temperature parameter reflecting the sensitivity of choice to the difference of values $V_i$. The decision variable or value $V$ of an option is modeled as the expected (average) reward + expected uncertainty + U-turn cost[19,33].

As mice could not receive two consecutive rewards on the same location, a $6 \times 3$ matrix is sufficient to describe the probability of choices between A, B, and C (the three targets) depending on the two preceding choices. For instance, after performing the sequence BA, the values for the three following options {A, B, C} are given by { $V_A = 0$; $V_B = p_b + \varphi * p_{b*}(1 - p_b) - \kappa$ ; $V_C = p_c + \varphi * p_{c*}(1 - p_c)$ }. The U-turn cost is only applied to the choice B, as the BAB sequence would constitute a U-turn. Likewise, after the sequence CA, the values are given by { $V_A = 0$; $V_B = p_b + \varphi * p_{b*}(1 - p_b)$ ; $V_C = p_c + \varphi * p_{c*}(1 - p_c) - \kappa$ }. The same holds after AB, CB, AC, BC sequences, effectively resulting in a $6 \times 3$ matrix of choices. The free parameters of the model ($\beta$, $\varphi$, $\kappa$) were fitted by maximizing the data likelihood. Given a sequence of choice $c = c_{1..T}$, data likelihood is the product of their

probability given by the softmax choice rule[68]. We used the optim function in R to perform the fits, with the constraints that $\beta \in ]0,10]$, $\varphi \in ]0,5]$ and $\kappa \in ]0,5]$.

In the model, the $\beta$ parameter reflects how much the difference in total value between the two options ($\Delta V$) translates into more or less preference for the best option in a given gamble. With a small $\beta$, choices have low sensitivity for $\Delta V$, with the extreme case of a null $\beta$ where both options have the same probability to be selected in each gamble ($G_{100} = G_{50} = G_{25} = 50\%$), leading to equal global distribution of visits ($P_{100} = P_{50} = P_{25} = 33\%$), independently of their respective value. On the contrary, a large $\beta$ indicates a high sensitivity to $\Delta V$, with an infinite beta indicating that options associated with higher reward probabilities are always selected ($G_{100} = G_{50} = 100\%$ and $G_{25}$ would not even exist considering that animals would never visit $P_{25}$, with $P_{100} = P_{50} = 50\%$ and $P_{25} = 0\%$).

*Model comparison.* To compare models[68], we used the Bayesian information criterion (BIC) to correct the raw likelihoods for the number of free parameters fit. BIC scores were aggregated across mice (Fig. 4b). M1 and M2 are nested cases of M3. In M3, $\beta > 0$, $\varphi > 0$, and $\kappa > 0$. In M1, $\beta$ and $\varphi = 0$, so that choices are only driven by a motor cost $\kappa > 0$. In M2, $\varphi = 0$, corresponding to animals that do not take uncertainty into account. A likelihood ratio test was used to estimate the probability of the observed data under the null hypothesis that these data are generated by the simplest model. For that we computed d, twice the difference in log likelihoods of M2 or M1 with M3. The probability of a significant difference d follows a chi-square distribution with a number of degrees of freedom n equal to the difference of parameters number between M3 and M1 or M2 (here $n = 1$ or 2)[68].

*Statistical analysis.* All statistical analyses were computed using R (The R Project, version 4.0.0) and Python with custom programs. The results were plotted as a mean ± sem. The total number ($n$) of observations in each group and the statistics used are indicated in figure legends. Classical comparisons between means were performed using parametric tests (Student's *t*-test, or ANOVA for comparing more than two groups when parameters followed a normal distribution (Shapiro test $P > 0.05$), and non-parametric tests (here, Wilcoxon or Mann-Whitney) when the distribution was skewed. Multiple comparisons were corrected using a sequentially rejective multiple test procedure (Holm). Probability distributions were compared using the Kolmogorov–Smirnov (KS) test, and proportions were evaluated using a chi-squared test ($\chi^2$). All statistical tests were two-sided except for the optogenetic experiment (Fig. 5) where statistical tests were paired and one-sided (we test hypotheses driven by nicotine effect and model). $P > 0.05$ was considered not to be statistically significant. For archetypal analysis, all computations and graphics have been done using the statistical software R and the archetype package (version 2.2-0.1). Briefly, given an $n \times m$ matrix representing a multivariate dataset with $n$ observations ($n$ = number of animals) and $m$ attributes (here $m = 7$, consisting of the directional changes rate, the target distributions (3 values) and the three gambles (see data, Fig. 1c–e)), the archetypal analysis finds the matrix Z of k m-dimensional archetypes ($k$ is the number of archetypes). Z is obtained by minimizing $|| X - \alpha Z^T ||_2$, with $\alpha$ the coefficients of the archetypes ($\alpha_{i,1..k} \geq 0$ and $\sum \alpha_{i,1..k} = 1$), and $||.||_2$ a matrix norm. The archetype is also a convex combination of the data points $Z = X^T\delta$, with $\delta \geq 0$ and their sum must be 1[69]. The $\alpha$-coefficient depicts the relative archetypal composition of a given observation. For $k = 3$ archetypes and an observation i, $\alpha_{i,1}$; $\alpha_{i,2}$; $\alpha_{i,3} \geq 0$ and $\alpha_{i,1} + \alpha_{i,2} + \alpha_{i,3} = 1$. A ternary plot can then be used to visualize data ($\alpha_{i,1}$, $\alpha_{i,2}$, $\alpha_{i,2}$) are used to assign individual behavior to its nearest archetype (i.e., $k$ max($\alpha_{i,1}$; $\alpha_{i,2}$; $\alpha_{i,3}$)). $\alpha_{i,j}$ are also used as variable to estimate population archetypal composition. For Fig. 4e, archetypal composition ($0 \leq \alpha_{i,j} \leq 1$) was binned into five intervals. Pure archetype corresponds to 1, the archetypal composition decreases linearly with increasing distance from the archetype, 0 correspond to points on the opposite side.

**Statistics and reproducibility.** All experiments were replicated with success.

**Reporting summary.** Further information on research design is available in the Nature Research Reporting Summary linked to this article.

## Data availability

The raw data from the online behavioral experiment (i.e., the trajectories) are available from the corresponding author. Source data are provided with this paper.

## Code availability

Animal's trajectories are collected with homemade LabVIEW program (version 2014). The results were generated using code written in R (version 4.1.0) and Python (version 3.8.5). All codes used to run the analysis are available from the authors upon request. A sample code for the model and archetype is publicly available at https://zenodo.org/record/5596424#.YXfBAy8ivgY

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

## Acknowledgements

We are grateful to the animal facilities (IBPS), Camille Robert and Paris Vision Institute AAV production facility for viral production and purification. This work was supported by the Centre National de la Recherche Scientifique CNRS UMR 8246, INSERM U1130, the Foundation for Medical Research (FRM, Equipe FRM DEQ2013326488 to P.F.), FRM FDT201904008060 (to S.M.), the French National Cancer Institute Grant TABAC-16-022 et TABAC-19-020 (to P.F.), French state funds managed by the ANR (ANR-16 Nicostress, ANR -17 SNP-Nic, ANR-20 Nicado to P.F., ANR-19 Vampire to F.M.) and The LabEx Bio-Psy (to P.F.). M.L.D., R.D.C., and S.M. were the recipients of a fourth-year PhD fellowship from FRM (FDT20160435171, FDT20170437427, and FDT201904008060), C.N. was recipient of a doctoral fellowship from the Labex Bio-Psy, D.L. was recipient of a post-doctoral Fellowship from the Labex Bio-Psy, and L.M.R. was supported by a NIDA–Inserm Postdoctoral Drug Abuse Research Fellowship.

## Author contributions

M.D., R.D.C., C.N. and M.C. contributed equally to this work. P.F. and M.D. designed the study. M.D., R.D.C., C.N., M.C., T.A.Y., E.K.D., R.B., E.B., B.H. and N.T. performed the behavioral experiments. M.D., R.D.C., S.M. and N.T. performed the minipumps implantations. J.N., D.L. and S.D. contributed to setup developments. C.N., S.M., R.D.C., D.L. and F.M. performed electrophysiological recordings. M.D., R.D.C., C.N., M.C., E.K.D., R.B., T.A.Y., E.B., N.T. and J.N. performed the surgeries and virus injections. C.N. and S.M. performed the immunohistochemistry experiments. D.D. provided the viruses. J.N. and P.F. developed the model. AM developed the optogenetic setup. M.D., R.D.C., C.N., M.C., S.M., J.N., F.M. and P.F. analyzed the data. P.F. wrote the paper with inputs from M.D., R.D.C., C.N., M.C., L.M.R., J.N., F.M. and A.M.

## Competing interests

The authors declare no competing interests.
