## [Peer Review File. · Nature Communications]

Chronic nicotine increases midbrain dopamine neuron activity and biases individual strategies towards reduced exploration in miceREVIEWER COMMENTS

Reviewer #1 (Remarks to the Author):

The manuscript by Dongelmans, Durand-de Cuttoli, Nguyen, Comme et al., outlines the effects of chronic nicotine exposure on exploratory/reinforcement driven behavior. One important aspect of drug use (including nicotine) is that it leads to changes in decision-making and neural function that are not related to drug-associated behaviors. These cognitive and behavioral deficits are an understudied aspect of addiction and in this regard the manuscript is timely and important. One of the major strengths of the manuscript is that it combines complex behavioral analysis, physiology, and neural manipulations to link the VTA to these behaviors. While the manuscript is generally well-presented and the findings are interesting, the manuscript would benefit from a more precise discussion of the data, and more explicit interpretation of some of the findings. I have listed my concerns below.

Major:

1. I am concerned about the terminology that the authors use to describe their results in many places throughout the manuscript. One of the major things highlighted is the neural mechanisms of suboptimal choice. In the legend to figure 1 they define the behavior as suboptimal, but it looks like the behavior is optimal based on the task parameters. The biggest issue is that they use probabilities about choice to make conclusions; however, the probabilities are not calculated correctly as they seem to be based on a situation of free choice – which this task is not.

One example - “the probability to choose p100 over p50 was not different from a random choice (Figure 1F), which has been interpreted as indicating that mice assign a positive motivational value to expected uncertainty” – I disagree with this statement. The choice was this way because it was not free choice and the experimenters put constraints on the task that made the animals behave in this fashion. If the animals were performing this task optimally the behavior should be exactly what the authors saw. If they cannot get two reinforcers in a row the most optimal behavioral performance would be to go back and forth between the p100 and p50. This would maximize the reinforcer delivery. This does not mean that the mice find them equally reinforcing or that they find uncertainty motivating (maybe that is the case, but these data do not show that), just that the task parameters made it so that they had to choose another option. If you made the task 100, 25, and 10, the mice would likely go back and forth between the 25 and the 100. This is important as the authors suggest that some of their data is looking at suboptimal choice and I do not think that have shown that here.

These issues are present throughout the manuscript and in my opinion paint a confusing picture of how the authors are interpreting their data.

2. Exploration and exploitation should be operationally defined. I am not sure what they mean in this context or how the authors know that that is exactly what they are testing. A lot of the manuscript focuses on “exploration/exploitation trade-off” which is fine to discuss but is impossible to tell if that is what exact behavior they are talking about. How do the authors know that these are specific strategies and not that some animals understand the task and some animals do not?

3. Along a similar line, the authors should consider being more careful with their wording. The idea that nicotine reduces exploration contrasts with data specifically showing the opposite in general contexts. I do agree that nicotine increases performance on this reinforcement task and that the strategy changes across animals depending on experience. It is hard to say if they are just better at making decisions, better at learning, or are exploring less in general.

4. The same could be said for the computational strategy – how do you know that the animals that do not fall into the GO phenotype understand the task parameters? Are these animals exploring? Or do they just not understand the task and are just going in a circle to each point and essentially are reinforced on a random ratio for running in a circle from point to point? This is incredibly important to the conclusions of the manuscript. It needs to be shown that the animals both understand the task and are just using different strategies, rather than the alternative where some animals have acquired, and some have not, and nicotine enhances learning.

5. Another major concern with the neural data is what the optogenetic studies mean. The authors seem to think that it suggests that stimulation of the VTA leads to VTA plasticity which drives these effects; however, if these changes were all in the VTA, wouldn't the MFB stimulation override this type of plasticity? It seems more likely that these changes are downstream of the VTA and not the VTA itself. As is, it is unclear to me that the VTA itself is driving these effects. It is just as likely

that nicotinic receptors in the NAc underlie many of these effects and stimulating the VTA repeatedly causes the same type of plasticity downstream through completely different mechanisms. I would just suggest softening the wording to allow for other possible conclusions that could also explain some of the data.

Minor

1. When talking about the physiology experiments the authors note that there are increases in tonic firing at baseline and during bursts, but bursts are typically thought of as phasic events. The manuscript would be strengthened by talking about the relationship between those two things, especially since nicotine is known to alter the relationship between tonic and phasic activity.

Reviewer #2 (Remarks to the Author):

This work comes from an excellent group of researchers that have a long-established history in the nicotine-related field. The noteworthy results in this study is the identification of nicotine's role in key aspects of motivation-related behaviors. This paper nicely shows that nicotine alters motivation to target high-reward at the loss of exploratory behavior. This work will be significant to the field of drug abuse as this provides more evidence that strong reward/reinforcers alter motivation toward specific goal-directed objectives. While the authors present an impressive set of data there are some key issues that need to be resolved. Namely, there is a lack of detail regarding statistical tests on a key experiment and this impedes the determination if there is strong evidence for their conclusions.

Major Comments

1. To facilitate a better comparison between the nicotine-treated condition (Figure 3G) to the optogenetic confirmation (Figure 5F) similar plot styles should be used. At present, the ternary archetype plot of 5F suggests there may only be a modest change toward GO behavior with optogenetic stimulation 'on'. Finally, the biggest problem with this data (5F) is there seems to be no clear description of the statistics used to state the difference. B, D and E all have descriptions of the statistics but the key experiment (5F) has no stats description (in legend or results section). Without this, one cannot adequately assess how impactful this key experiment is.
2. This study is done completely with male mice. The authors should provide a justification for using males and discuss what findings may be different if females were included in the study. Alternatively, if the authors are currently pursuing this work with females, they can briefly mention that.
3. Given that in vivo and ex vivo electrophysiology is used in this work, the authors need to clarify this in the results section. For instance, in describing the experiments of Figure 2B, there is no clear dissemination that this is in vivo recordings from VTA Da neurons. While experienced physiologists will understand this from the figure many in the behavior/neuroscience field may not pick up on this. This is important given that ex vivo electrophysiology of VTA dopamine neurons have shown numerous times that nicotine treatment decrease dopamine firing but this is due to the lack of excitatory projects that are removed due to coronal slice preparation. Thus, it needs to be made clear which recordings are in vivo recordings and which are ex vivo.
4. Given that other drugs of abuse may trigger similar mechanisms, the authors should discuss how cocaine, opioids, etc. may trigger similar changes in motivated behavior and how particular aspects may differ with other reinforcers. If the authors think this is a phenomenon unique to nicotine, they should discuss that instead.

Minor Issues to improve rigor and clarity

1. The authors should clarify if the freebase weight of nicotine was corrected for in making their solutions for use with osmotic minipumps.
2. If a commercial bipolar stimulating electrode was used, the vendor/model number should be provided
3. The authors should provide a sample Jaws off/on trace to Figure 5C

Reviewer #3 (Remarks to the Author):

This study asks whether exposure to nicotine alters the explore/exploit balance, and uses nicotine administration and then optogenetic stimulation to test behavior of mice performing a stochastic three-arm bandit task. The key findings are that nicotine increases exploitation and that DA enhancement via optogenetics does the same.

I don't know anything about these methods or species, so I will refrain from commenting on them. Instead I will focus on conceptual issues.

The major limitation of this paper in my view is that the broader research goal is underspecified - like, why do this study? Clearly the authors are interested in understanding the psychological effects of chronic nicotine exposure. However, the reason they want to understand explore/exploit behavior is not given. The explore/exploit tradeoff (which I do know some things about), is kind of a weird focus. Like, is the goal to develop cognitive treatments for smoking? Or to predict smoking susceptibility in non-smokers? I wasn't sure.

More importantly, explore/exploit behavior reflects a whole bunch of underlying psychological processes. These include, for example, curiosity, motivation, desire for reward, impulsivity, capacity for learning, and attentional state. There are several more. The design of the study does not really allow the authors to disambiguate these.

So, for example, the subjects could become better at the task because of well-established effects of nicotine, such as increased working memory or ability to learn. Or they could be due to other less well studied factors, such as "learning rate/temperature". The authors do link behavioral change to changed reward sensitivity but (1) this is already established for nicotine, and (2) it doesn't exclude other processes, nor does it suggest that this is the largest factor.

So, what, in the end, have we learned? This study uses a complex behavioral readout that touches on nearly all aspects of cognition, and link nicotine to that readout, but without going much beyond that. Ultimately, I am not sure there is much solid that we can take away from the study.

Relatedly, the explore/exploit balance is not really a specific reified thing; instead, it is the outcome of a large number of psychological processes. That's also true of many other complex multifactorial behaviors - so why is explore/exploit interesting?

L341 - is "exploration" really a "central personality trait"? I've never heard that idea expressed before. Nor am I convinced that behavior in a bandit task can measure personality.

Text would be easier to read if paragraphs were indented.

Why are PS and DS abbreviated? That just makes the paper more confusing to read. Likewise there is absolutely no reason to abbreviate control as "Ctl" - why not just say "control"? Same with "nic" for "nicotine".

REVIEWER COMMENTS

Reviewer #1 (Remarks to the Author):

The manuscript by Dongelmans, Durand-de Cuttoli, Nguyen, Comme et al., outlines the effects of chronic nicotine exposure on exploratory/reinforcement driven behavior. One important aspect of drug use (including nicotine) is that it leads to changes in decision-making and neural function that are not related to drug-associated behaviors. These cognitive and behavioral deficits are an understudied aspect of addiction and in this regard the manuscript is timely and important. One of the major strengths of the manuscript is that it combines complex behavioral analysis, physiology, and neural manipulations to link the VTA to these behaviors. While the manuscript is generally well-presented and the findings are interesting, the manuscript would benefit from a more precise discussion of the data, and more explicit interpretation of some of the findings. I have listed my concerns below.

We thank the reviewer for her/his positive comments. We hope we have understood his/her concerns and have changed the text accordingly to make it clearer and avoid misunderstandings. We have added new analysis and data to answer her/his questions. Overall, i) we clarify our interpretation of a choice behavior as suboptimal and our operational definition of the exploit/explore tradeoff, ii) we show how model comparison excludes alternative interpretations (e.g. that some animals do not understand the task or learning effects), and iii) we show stability of individual strategies. We now also discuss physiology and optogenetics points raised by Reviewer #1.

Major:

1. I am concerned about the terminology that the authors use to describe their results in many places throughout the manuscript. One of the major things highlighted is the neural mechanisms of suboptimal choice. In the legend to figure 1 they define the behavior as suboptimal, but it looks like the behavior is optimal based on the task parameters. The biggest issue is that they use probabilities about choice to make conclusions; however, the probabilities are not calculated correctly as they seem to be based on a situation of free choice – which this task is not. One example - “the probability to choose p100 over p50 was not different from a random choice (Figure 1F), which has been interpreted as indicating that mice assign a positive motivational value to expected uncertainty” – I disagree with this statement. The choice was this way because it was not free choice and the experimenters put constraints on the task that made the animals behave in this fashion. If the animals were performing this task optimally the behavior should be exactly what the authors saw. If they cannot get two reinforcers in a row the most optimal behavioral performance would be to go back and forth between the p100 and p50. This would maximize the reinforcer delivery. This does not mean that the mice find them equally reinforcing or that they find uncertainty motivating (maybe that is the case, but these data do not show that), just that the task parameters made it so that they had to choose another option. If you made the task 100, 25, and 10, the mice would likely go back and forth between the 25 and the 100. This is important as the authors suggest that some of their data is looking at suboptimal choice and I do not think that have shown that here. These issues are present throughout the manuscript and in my opinion paint a confusing picture of how the authors are interpreting their data.

It seems that Reviewer #1 is confusing the global repartition of visits to the three targets (Figure 1e) with the probability to choose the option with the highest probability of reward in each of the three binary gambles (Figure 1f). We now state more clearly in the results that the global repartition of visits cannot be directly interpreted as free choice, but constitutes the outcome of successive binary choices, each of them being made in one of three situations of free choices between pairs of reward probabilities (e.g. gambles).

We agree with the referee that if we consider the global repartition of visits (as in Figure 1e), the proportions of visits p_{100} and p_{50} would be the same in an optimal scheme, and this reflects constraints on the task: mainly the fact that mice could not receive two consecutive rewards from the same target. In such optimal scheme, the mice would indeed go back and forth between p100 and p50 to optimize reward success rate and the proportion of visits to the three targets would then be $p_{100}=50\%$, $p_{50}=50\%$ and $p_{25}=0\%$. Along with the same idea, if we made the task 100, 25, and 10, the reviewer is right, mice would likely go back and forth between the 25 and the 100, while visiting 10 would be sub-optimal.

However, when we stated “the probability to choose p100 over p50 was not different from a random choice (Figure 1f)” we referred to the probability to choose p100 over p50 when the animal starts from p25, i.e. in a specific gamble,

labelled G_{25} in Figure 1f, corresponding to a free choice between two reward probabilities. In this case, when an animal's last visit is p_{25} , so that it is now confronted with a binary choice between the two targets with a probability of 100% or a probability of 50% to receive a reward respectively, one could expect that it would choose the 100% target more often. If the animal only exploits, it would even always prefer the 100% target in such situation. This was not the case in the data: mice chose equally between 100% and 50% (Figure 1f, middle bar of the Barplot corresponding to gamble G_{25}).

The analysis of the probabilities to choose the option with the highest probability of reward for each of the three possible gambles is at the core of this article. Not only does this analysis point towards what can be called suboptimal behavior (for example choose equally between 100% and 50% when coming from the 25% target as mentioned before, but also the fact that mice do not always prefer 100% over 25% when coming from the 50% target as shown in Figure 1f, right bar of the Barplot corresponding to gamble G_{50}), but it is also the basic data used for fitting and simulating the model. Indeed, i) the number of each choice in the three gambles is used to fit the data and to estimate (β , φ , κ) parameters and ii) the simulations are based on sequential binary choices as performed by the animals.

To avoid possible sources of confusion, we now state in the Results (line 109-113): *"While a purely exploitative strategy would consist solely in an alternation of visits between p_{100} and p_{50} , we found that mice continued to visit all three points, prompting us to further investigate the exploration/exploitation tradeoff in their choices. However, the global repartition of visits does not directly measure choices. Indeed, since mice could not receive two consecutive rewards from the same target, the repartition of visits on the three locations resulted from a sequence of binary choices (Figure 1f)..."*

2. Exploration and exploitation should be operationally defined. I am not sure what they mean in this context or how the authors know that that is exactly what they are testing. A lot of the manuscript focuses on "exploration/exploitation trade-off" which is fine to discuss but is impossible to tell if that is what exact behavior they are talking about.

We understand the concerns about exploration/exploitation wording raised by the reviewer. Standard in decision-making studies, the exploration and exploitation are operationally defined in the context of the choice in each gamble. When faced with a choice between two alternatives, exploitation corresponds to choosing the option for which the animal assigns the highest value, while exploration corresponds to choosing the less valued alternative. Animals purely exploiting will always choose the high probability option. But in our setup, animals choose the less likely rewarded option a significant portion of the time. This is consistent with animals intermixing exploitation and exploration. By measuring, for each free choice ("gamble"), the distance to pure exploitation, we can thus estimate the degree by which the animals explore or exploit. The SoftMax model of decision-making is designed to quantify this degree of exploitation/exploration through the β parameter. "A small β favors exploration (the proportion of respective choices is less sensitive to ΔV , with a null β meaning all options have nearly the same probability to be selected, independently of their respective value), while a large β indicates exploitation (high sensitivity to ΔV , with an infinite β meaning that options associated with higher reward probabilities are always selected)" (line 237). This is how we operationally defined the tradeoff between exploration and exploitation in our experiment. To avoid confusion, we now state clearly that we do not measure exploitation nor exploration per se, but their tradeoff, and we provide both a conceptual definition in the introduction (lines 48-65), and an operational definition based on free choices (gambles) in the first part of the Results (line 109 ; line 115).

How do we know that these are specific strategies and not that some animals understand the task and some animals do not?

To address Reviewer #1's question, we tested whether individual data could be explained by differences in learning stages, differences in understanding of the task, or actual differences in strategy. We add new panel (Figure 4b) and two new supplementary figure to address reviewers' questions (Supp Figure 4 and 5)

First, the design of the task excludes explanations based on learning. Indeed, despite significant adaptation of decision-making at the transition between deterministic (DS) and probabilistic setting (PS) and during the first half of PS (learning phase), the animals' behavior eventually becomes stable and the choice behavior is at steady-state

at the end of PS. This is demonstrated by the convergence to a stable value after 8-10 sessions for several of our choice parameters, and also, from session 6 until the end of PS, by the absence of positive correlation between the session number and the behavioral parameters (this is now stated Supp Figure 4, and line 195). Thus, with more time to learn, animals do not progress to the GO archetype.

As differences in learning are excluded, we next tested whether the observed diversity of behaviors is better modeled by some animals not understanding the task. We translated the hypothesis that mice did not understand the task with a model “M1” in which all points have the same value, as if mice considered a random, equal reinforcement on all points). In this model choices are only driven by one motor cost parameter (i.e. a bias against U-turn, explaining circling behaviors like in DS). Our principal model (already present in the last version), now called M3, considers three free parameters (β , ϕ , and κ for value sensitivity, uncertainty bonus and motor cost, respectively). We also introduced an intermediate M2 model considering motor cost (κ) and value sensitivity (β), but not uncertainty bonus (ϕ). Comparison of the three models is now shown in Figure 4b and in a new Supplemental Figure 5. We used Bayesian information criteria (likelihood penalized by number of model parameters) and likelihood ratio test for nested models (M1 and M2 are special cases of M3, Daw 2011 ref 68) to estimate the probability (per individual) of the observed data under the null hypothesis that the data are generated by the simpler model. Overall, all the control mice except one showed a positive difference. This positive difference indicates that the principal model M3 explains better the behavior than the others (M1 or M2), for all 55 mice except this one. In particular, this statistical test rejects the M1 (cost-only) model as a likely explanation for mice behavior. Hence, the M3 model, which considers that mice only differ in their strategies (i.e. their decision-making parameters), explains better the experimental data.

We now detail in the Results (*line 252*) the model comparison, and how it suggests that all mice do understand that the three locations are associated with different reward probabilities, and are also sensitive to reward uncertainty. Result of model comparison is also detailed Figure 4b and in Supplementary Fig 5.

3. Along a similar line, the authors should consider being more careful with their wording. The idea that nicotine reduces exploration contrasts with data specifically showing the opposite in general contexts. I do agree that nicotine increases performance on this reinforcement task and that the strategy changes across animals depending on experience. It is hard to say if they are just better at making decisions, better at learning, or are exploring less in general.

In the context of decision-making, nicotine has been suggested to reduce exploration using bandit tasks in humans, and we are not aware of data showing opposite effects.

Moreover, similar to control groups, the model comparison allows to infer the task parameters that are the most likely to affect the choices under nicotine. As stated above, animals most likely behave as if they are sensitive to *i*) expected rewards, *ii*) the uncertainty associated with each choice and *iii*) the motor cost of performing U-turns. As already mentioned in the answer to point 2, the β parameter reflects how much the difference in total value between the two options (ΔV) translates into more or less preference for the best option in a given gamble. With a small β , choices are less sensitive to ΔV , with a null beta meaning both options have nearly the same probability to be selected in each gamble ($G_{100} = G_{50} = G_{25} = 50\%$), leading to equal global distribution of visits ($p_{100} = p_{50} = p_{25} = 33\%$), independently of their respective value. On the contrary, a large β indicates a high sensitivity to ΔV , with an infinite beta meaning that options associated with higher reward probabilities are always selected ($G_{100} = G_{50} = 100\%$ and G_{25} would not even exist considering that animals would never visit the 25% location, with $p_{100} = p_{50} = 50\%$ and $p_{25} = 0\%$). This is explained in the Methods section (*Modeling, Line 616*). High β would then reflect high exploitation, while low beta would reflect maximal exploration. In this sense, β can thus be considered as a proxy to measure the exploration/exploitation tradeoff and being « better at making optimal decisions » and « explore less » are thus equivalent in the exploit-explore framework. This is explained in the Results (*line 240: β can thus be considered as a proxy to measure the exploration/exploitation tradeoff. “Choosing the highest rewarded option” and “exploring less” are therefore equivalent in this exploit/explore framework*).

Finally, as stated in the previous answer, it is unlikely that the observed difference stems from better learning. After about ten sessions, the choice behavior of control mice is stable (Figure Supp 4) hence if nicotine effect was to

speed up learning, this steady-state would only occur earlier in the learning sessions, but the value reached for a given parameter when the animal is at steady-state would not change. A related argument is given by optogenetic results, as optogenetic stimulation of the VTA exerts an immediate, within session with successive ON and OFF sessions (hence not attributable to learning) effect on decisions. Since this optogenetic manipulation recapitulates nicotine effects on choices, it adds evidence in favor of a strategy or policy, rather than learning, effect of nicotine.

4. The same could be said for the computational strategy – how do you know that the animals that do not fall into the GO phenotype understand the task parameters? Are these animals exploring? Or do they just not understand the task and are just going in a circle to each point and essentially are reinforced on a random ratio for running in a circle from point to point? This is incredibly important to the conclusions of the manuscript. It needs to be shown that the animals both understand the task and are just using different strategies, rather than the alternative where some animals have acquired, and some have not, and nicotine enhances learning.

First, considering all mice to have the potential to become gain-optimizers in the task but that some simply did not have enough time to learn is unlikely. Choice behavior was stable along last sessions (see answer 3 above and Supp. Fig. 4). Moreover, the absence of correlation between sessions and model parameters during the second half of PS indicates that, with more time to learn, animals closer to Asc or Des archetypes do not progress to the GO archetype. So, at the end of the PS, since mice are trained and choice behavior is at steady-state, suggesting that all mice have the same knowledge about the environment (the estimation of the probability, and associated uncertainty) and are not in transition towards the GO phenotypes; they simply don't use knowledge about the statistical structure of the environment the same way as mice closer to the GO archetype (the way mice use this knowledge to influence their choices, which is called strategy or policy, is represented by the variation in the parameters of the model). This is now clearly stated line 225 and Figure Supp 4: L 195: *“Inter-individual variabilities of choice patterns in mice were not the consequence of random variations or different stages in the same learning process. Rather, they were robust in each animal throughout the task, indicating differences in individual strategies. This is suggested first by the overall stability of the behaviors as indicated by the convergence to a plateau at sessions 8-10 (Supplementary Figure 4a), and by the absence of any positive correlations between decision-making parameters and session number after the first 5 sessions (Supplementary Figure 4b). Furthermore, to test whether the variabilities in behavior were robust for each individual from trial to trial, we compared the percentage of directional changes for two consecutive sessions for each animal of the control group. Directional changes showed a strong positive correlation from one session to the next (Figure 3c), suggesting a strong consistency in individual behaviors. This observation was generalized by demonstrating that intra-individual variations are lower than the inter-individual variations (Supplementary Figure 4c).”*

Second, considering that some mice just did not understand the task is also unlikely. In the model comparison (see answer 3 above), the model M1 corresponds to agents “just going in a circle to each point” and “reinforced on a random ratio”. The behavior of control mice (except one) is better explained by a model considering that animals do assign different values to the three points. This indicates that animals understand the task. Moreover, the model fit provides a direct measure of differences in strategy (especially β) for which we display all individual data (Figure 4B) and variation in the parameters value depending on the archetypal composition (Figure 4f).

Last, the optogenetics experiments rapidly altered the behavioral strategies of the mice, with kinetics that are incompatible with synaptic plasticity or learning. This suggests that animals understand the task and that their DA tone influences how they use information about the statistical structure of the environment.

We also now add a specific part on that topic in the discussion (see L363-386).

5. Another major concern with the neural data is what the optogenetic studies mean. The authors seem to think that it suggests that stimulation of the VTA leads to VTA plasticity which drives these effects; however, if these changes were all in the VTA, wouldn't the MFB stimulation override this type of plasticity? It seems more likely that these changes are downstream of the VTA and not the VTA itself. As is, it is unclear to me that the VTA itself is driving these effects. It is just as likely that nicotinic receptors in the NAc underlie many of these effects and stimulating the VTA repeatedly causes the same type of plasticity downstream through completely different mechanisms. I would just suggest softening the wording to allow for other possible conclusions that could also explain some of the data.

Optogenetic stimulation of the VTA exerts an immediate effect, within one session and once the mice display steady-state choice behavior in the probabilistic setting. We thus do not imply in the manuscript that VTA photostimulation exerts plasticity effects downstream. In contrast, we agree with Reviewer #1, and think that both nicotine (which among many effects on other structures, elevate the firing rate of VTA DA neurons, see Figure 2b), and VTA photostimulation (Figure 5), likely changed the downstream DA release and consequently affected the way mice make decisions (i.e. their strategy or policy) rather than the way they learn. It is clear that MFB stimulation exerts a plastic and reinforcing effect throughout the 10 DS sessions to condition mice to the task, as well as in the PS 10 sessions to affect the respective values mice assign to each of the three locations. However, as optogenetics are used at the end of PS, once mice have learned the association between each location and the reward probability and are at steady-state in terms of choice behavior, it is unlikely that optogenetics-induced elevation of VTA DA neurons firing affect decision making parameters through learning effects, nor do we believe that MFB-induced long-term plasticity would affect (e.g. override or other interactions) the immediate, acute effects of VTA photostimulation in the last sessions. On the contrary, we proposed that changes in VTA DA firing and in DA release in target structures dynamically regulate the exploration/exploitation tradeoff, through classical neuromodulation. This is different and independent to the learning effect mediated by the MFB stimulation. This is now better outlined in the discussion (*line 346 and line 406*).

Minor

1. When talking about the physiology experiments the authors note that there are increases in tonic firing at baseline and during bursts, but bursts are typically thought of as phasic events. The manuscript would be strengthened by talking about the relationship between those two things, especially since nicotine is known to alter the relationship between tonic and phasic activity.

We thank the Reviewer #1, it is indeed a good opportunity to distinguish bursting from phasic activities, two concepts that derive from two different contexts. In the context of dopamine neuron physiology, bursting has been defined as successive action potentials separated by less than 80ms (*Grace and Bunney 1984, ref 48*) occurring on top of their regular “pacemaking” firing activity. In this context, bursting is not necessarily locked to any behaviorally relevant or salient event. By contrast, phasic activity is related to event-locked increase in firing (*Schultz 2007, ref 22*), which can typically be observed as an increased density in peri-stimulus histograms, but are not necessarily composed of bursts of action potentials (i.e. it can be single spikes but time-locked to an event during successive trials). This is now discussed in the discussion (*lines 398-405*). In our experiment, we recorded in-vivo spontaneous electrophysiological activities of VTA DA cells in anesthetized animals (i.e. in a context independent of any stimuli), and we evaluated the average firing rate and % of spikes within bursts for each cell. An increase of this parameter suggests that dopaminergic tone (both spiking and bursting tone) is modified in nicotine-exposed animals, and both can bear consequences on behavioral expression. This is now also discussed (*line 411 and 416*).

Reviewer #2 (Remarks to the Author):

This work comes from an excellent group of researchers that have a long-established history in the nicotine-related field. The noteworthy results in this study is the identification of nicotine’s role in key aspects of motivation-related behaviors. This paper nicely shows that nicotine alters motivation to target high-reward at the loss of exploratory behavior. This work will be significant to the field of drug abuse as this provides more evidence that strong reward/reinforcers alter motivation toward specific goal-directed objectives. While the authors present an impressive set of data there are some key issues that need to be resolved. Namely, there is a lack of detail regarding statistical tests on a key experiment and this impedes the determination if there is strong evidence for their conclusions.

We thank the reviewer for her/his positive comments

Major Comments

1. To facilitate a better comparison between the nicotine-treated condition (Figure 3G) to the optogenetic confirmation (Figure 5F) similar plot styles should be used. At present, the ternary archetype plot of 5F suggests there may only be a modest change toward GO behavior with optogenetic stimulation ‘on’. Finally, the biggest

problem with this data (5F) is there seems to be no clear description of the statistics used to state the difference. B, D and E all have descriptions of the statistics but the key experiment (5F) has no stats description (in legend or results section). Without this, one cannot adequately assess how impactful this key experiment is.

We agree that the presentation of some results was not optimal. In Figure 5f we displayed differences between “On” and “Off” stimulation because it is paired data whereas Figure 3g presents unpaired data (control vs nicotine mice groups). We apologize for the missing statistics for 5f (only the asterisk was provided), the statistics are now given in the legends and text. We completely changed our presentation of the data and now provide the reader with individual paired ON and OFF data in the Figure 5 (for some of the key parameters) and in supplementary figure 6. Our statistical reasoning is organized as follows. We hypothesize that optogenetic activation of VTA DA neurons will reproduce the effect of nicotine on choice behavior, while optogenetic inhibition will produce the opposite effect (lines 305). For this reason, we test only the parameters that have been modified by nicotine and use unilateral testing (see Figure legend and methods). We also compare the net effect of nicotine, with that of photostimulation and of photoinhibition on eight parameters independently of the significance of nicotine effects (Supplementary Figure 6e). We then quantified the number of parameters that vary in the same direction as nicotine during photostimulation or photoinhibition, and used a chi-squared test to compare the distribution between the two groups (Supplementary Figure 6e).

2. This study is done completely with male mice. The authors should provide a justification for using males and discuss what findings may be different if females were included in the study. Alternatively, if the authors are currently pursuing this work with females, they can briefly mention that.

Our experiments have been done mainly in male mice, but the replication with females of the results obtained in the probabilistic setting (this paper and the Naudé et al 2016 paper) is work in progress. Currently, we do not see any differences between males and females in the deterministic setting, nor in the complexity task (from Belkaid et al. et al. 2020), although those data have not been published yet. We now note in the discussion that sex or strain differences may constitute another layer of variability, and that we pursue our work in this direction (*line 369: Note that sex or strain differences may constitute another layer of variability*⁴⁴, which we are currently addressing in ongoing experiments.)

3. Given that in vivo and ex vivo electrophysiology is used in this work, the authors need to clarify this in the results section. For instance, in describing the experiments of Figure 2B, there is no clear dissemination that this is in vivo recordings from VTA DA neurons. While experienced physiologists will understand this from the figure many in the behavior/neuroscience field may not pick up on this. This is important given that ex vivo electrophysiology of VTA dopamine neurons have shown numerous times that nicotine treatment decrease dopamine firing but this is due to the lack of excitatory projects that are removed due to coronal slice preparation. Thus, it needs to be made clear which recordings are in vivo recordings and which are ex vivo.

We thank Reviewer #2 and we now add a clearer description of the electrophysiology experiments in the Results. We also add schematics (Figure 2b and Figure 5) to clearly indicate type of electrophysiological experiments (in-vivo juxtacellular or slice recordings), and specify the type of electrophysiology in the figure legends.

4. Given that other drugs of abuse may trigger similar mechanisms, the authors should discuss how cocaine, opioids, etc. may trigger similar changes in motivated behavior and how particular aspects may differ with other reinforcers. If the authors think this is a phenomenon unique to nicotine, they should discuss that instead.

We agree that applications of the explore/exploit paradigm to other drugs of abuse is an important point to discuss (we now do, see line 521), even though animal studies in bandit tasks are often missing. (*lines 437 Finally, such an explore-exploit paradigm and archetypal analysis could be very useful to study the effects of other drugs of abuse on decision-making. Indeed, humans with methamphetamine*⁶¹ *or alcohol use disorders*⁶² *have been suggested to present alterations in bandit tasks, but human studies cannot disambiguate whether altered decision-making facilitates, or results from, drug use.*)

Minor Issues to improve rigor and clarity

1. The authors should clarify if the freebase weight of nicotine was corrected for in making their solutions for use with osmotic minipumps.

Indeed we corrected the nicotine tartrate solution to achieve a final concentration in free base. This is now noted on line 473 "at a dose of 10 mg/kg/day (free base)".

2. If a commercial bipolar stimulating electrode was used, the vendor/model number should be provided

For all mice conditioned in the ICSS foraging task, Plastics One bipolar electrodes, 2-channels, stainless steel, 10mm (Phymep) were used. This is now stated line 461 in the Methods.

3. The authors should provide a sample Jaws off/on trace to Figure 5C

This figure has been updated to include the requested information.

Reviewer #3 (Remarks to the Author):

This study asks whether exposure to nicotine alters the explore/exploit balance, and uses nicotine administration and then optogenetic stimulation to test behavior of mice performing a stochastic three-arm bandit task. The key findings are that nicotine increases exploitation and that DA enhancement via optogenetics does the same. I don't know anything about these methods or species, so I will refrain from commenting on them. Instead I will focus on conceptual issues.

The major limitation of this paper in my view is that the broader research goal is underspecified - like, why do this study? Clearly the authors are interested in understanding the psychological effects of chronic nicotine exposure. However, the reason they want to understand explore/exploit behavior is not given. The explore/exploit tradeoff (which I do know some things about), is kind of a weird focus. Like, is the goal to develop cognitive treatments for smoking? Or to predict smoking susceptibility in non-smokers? I wasn't sure.

Research on the pharmacological effects of nicotine cannot be limited to developing treatments against tobacco addiction. Nicotine, through its specific action on nicotinic receptors located not only in the VTA but also in many other regions of the brain, exerts a diversity of psychological effects, and it has also been investigated as a therapeutic treatment for a variety of neuropsychiatric conditions. To this extent, a broader understanding of nicotine effects in animal models is a necessary step to understand the underlying mechanisms of the drug action, at the molecular and circuit levels. Following the comments of Reviewer #3, we now clarify the rationale for our specific research goals in the Introduction (see lines 38 to 65).

More specifically, we agree that the concept of the explore/exploit tradeoff is relatively new to psychiatric research, but reinforcement learning theory is increasingly used in the context of psychiatry to dissect modifications in specific components of decision-making. Traits and behaviors linked to decision making (attribution of subjective value, value sensitivity and decision policy, etc...), that are affected by chronic nicotine exposure and that extend beyond common symptoms of addiction are important to understand the dynamics of nicotine user profiles. However, studies of human decision-making are mostly correlative, with causal links remaining open questions as highlighted by Reviewer #3 (does altered decision-making facilitate or result from drug use, or a combination of the two?). We and others (e.g. Addicott et al. *Neuropsychopharmacology*, 2017) believe that the exploration-exploitation tradeoff, by providing formal, quantitative measures of action selection, is an ecologically valid tool for translational research. Here we provide clear evidence for a causal link (and a tonic dopamine mechanism) from nicotine exposure to general alterations in decision-making.

Explore/exploit is of particular interest (as we now state more clearly in the Introduction) because it is poised to impact the global equilibrium of decisions between drug and non-drug rewards. In the literature, explore/exploit dilemma are often related to selecting unknown, uncertain or low value options which can still provide useful information about the environment in the long term to update one's subjective value representation of many available options in the everyday life, vs selecting the option leading to the highest reward that one has already experienced. If nicotine exposure increases exploitation, we could predict that chronic nicotine users will focus on the highest rewards available - including nicotine itself - rather than smaller but ethologically appropriate and/or

hypothetical future rewards. Hence the effect of nicotine on the explore/exploit trade-off would participate in drug addiction without being related to drug reinforcement *per se*. This may also lead to dramatic consequences – yet hard to identify and quantify – in their everyday life as soon as they face value-based decisions, as already mentioned at the end of the discussion.

We believe this study adds to our general understanding of the psychological effects of chronic nicotine, and have now clarified and strengthened this rationale throughout the manuscript (in the introduction (line 39, 57,60) or in the discussion (last paragraph))

More importantly, explore/exploit behavior reflects a whole bunch of underlying psychological processes. These include, for example, curiosity, motivation, desire for reward, impulsivity, capacity for learning, and attentional state. There are several more. The design of the study does not really allow the authors to disambiguate these.

We agree with Reviewer #3 that the explore/exploit trade-off in behavior is related to several psychological processes. However, we do believe that both the design of the experimental study and computational modeling allow us to disambiguate a number of them. First, and contrary to similar tasks in humans, in our setup mice behavior is at steady-state after ten sessions, so we can exclude novelty, as well as attention (values are learned through slow reinforcement learning spread over 2 weeks of daily 5min sessions, rather than through within-session working memory or learning effects (choices are measured at steady-state, at the end of probabilistic setting)). We also control for the number and duration of trials by adjusting the ICSS current intensity for each mouse during the learning phase of deterministic setting (typically, until around session 5) to stabilize them between 50 and 100 trials in 5min (corresponding to 3 to 6s trials on average) which is used as a common starting point. Then intensity is maintained the same for each mouse from the end of deterministic setting and throughout probabilistic setting, so we can also exclude differences in impulsivity or motivational biases between the groups before we expose them to Nicotine or to probabilistic setting. We thus agree it leaves, as potential underlying processes, curiosity and desire for reward. These are precisely the factors we model: the φ parameter assesses the sensitivity of choices to uncertainty (a basic form of curiosity) and the β parameter measures the sensitivity to value (with higher β corresponding to high desire for reward, and lower β corresponding to undirected exploration). We also test for motor control explanation using the model. In addition, we now include a comparison between progressively constrained versions of our original model showing that animals' choices are more likely to depend upon these three factors together (value of uncertainty, value sensitivity, motor cost) than any single one of these parameters alone (see answer 2 to Reviewer #1 above). We now state more clearly in the Results when presenting the model how the task design and model comparison disambiguate factors underlying the explore/exploit tradeoff (lines 252).

So, for example, the subjects could become better at the task because of well-established effects of nicotine, such as increased working memory or ability to learn. Or they could be due to other less well studied factors, such as "learning rate/temperature". The authors do link behavioral change to changed reward sensitivity but (1) this is already established for nicotine, and (2) it doesn't exclude other processes, nor does it suggest that this is the largest factor.

As indicated above, and now clarified in the manuscript (line 197, supplemental Figure 4), choice behavior is at steady-state at the end of probabilistic setting, so our nicotine data is unlikely due to effects on working memory, or on their ability to learn/rate of learning. The computational model suggests instead that the most likely explanation results from differences in the temperature parameter (β), which is mathematically equivalent to reward sensitivity. This is now discussed in the results (line 252 and discussion line 373).

We agree that nicotine exposure has been linked to a decrease in reward threshold (i.e. the minimal amount of reward required to work on a single option), but not to value sensitivity in the context of a choice between two options and the explore/exploit tradeoff. Moreover, the interpretation that our chronic nicotine exposure paradigm mainly acts on choice policy through an increased sensitivity to reward value is further supported by our optogenetics data. Indeed, acute excitatory photostimulation of VTA DA cells at the end of the probabilistic setting shows, with paired comparison on the same animals, a decrease in exploration (increased value sensitivity, β) mimicking the effect of chronic nicotine. This is now discussed line 406-415.

So, what, in the end, have we learned? This study uses a complex behavioral readout that touches on nearly all aspects of cognition, and link nicotine to that readout, but without going much beyond that. Ultimately, I am not sure there is much solid that we can take away from the study.

Relatedly, the explore/exploit balance is not really a specific reified thing; instead, it is the outcome of a large number of psychological processes. That's also true of many other complex multifactorial behaviors - so why is explore/exploit interesting?

This preclinical study provides solid insight on how nicotine alters choice behavior, and is intended to complement and mechanistically extend studies in humans, which are primarily correlative and thus limited in their interpretability. Our design is based on long-term reinforcement, and includes a simplified choice environment with mechanistic circuit dissection in order to isolate as much as possible the nicotine-induced changes underlying the explore/exploit trade-off and manipulate them. In addition, we used modeling to determine which task parameters the animals are sensitive to. Hence, after disambiguating alternative candidates, we show that nicotine alters the decision temperature (i.e. β , or sensitivity to ΔV). This is important because it suggests that chronic nicotine users are pushed to focus on the highest rewards available, including nicotine itself. We thus suggest (line 62, line 433) that the effect of nicotine on the explore/exploit balance pervades everyday-life decisions and participates in drug addiction without being related to drug reinforcement per se. Furthermore, we provide a mechanistic understanding at a neurocircuit level (role of dopamine tone). In addition, while the goal of this study is not necessarily to provide directly new avenues for nicotine cessation interventions, we cannot preclude the usefulness of these mechanistic insights in informing future treatments and/or studies targeted specifically at developing treatments.

L341 - is "exploration" really a "central personality trait"?? Ive never heard that idea expressed before. Nor am I convinced that behavior in a bandit task can measure personality.

We agree, a reference for this idea was missing, and is now added (ref 55 et 56, Smillie et al 2014, De young 2014). If we consider personality traits as stable, consistent responses to broad classes of stimuli, then exploration, as the response to novel, uncertain or low value options, indeed constitutes a central personality trait, which means that its relationship with dopamine is indeed of broad interest to the community. Besides, the goal of this article was not to measure personality in the broad sense but to quantify one of the facets of decision-making in an ecologically valid task, and further identify archetypal strategies, which includes an individual's propensity to explore, among interindividual variations.

Text would be easier to read if paragraphs were indented. Why are PS and DS abbreviated? That just makes the paper more confusing to read. Likewise there is absolutely not reason to abbreviate control as "Ctl" - why not just say "control"? Same with "nic" for "nicotine".

All changes have been done accordingly.

REVIEWERS' COMMENTS

Reviewer #1 (Remarks to the Author):

The authors have addressed most of my concerns.

My one remaining suggestion is to go through the manuscript and make the text easier to follow for someone that is not directly in the field. The findings are interesting, but will have much more impact if they are presented in a way that is more accessible.

Reviewer #2 (Remarks to the Author):

This current version of the manuscript has been improved by the authors in many ways. As stated in my previous review, this work is impactful because it provides new observations regarding nicotine's ability to alter motivation-related behavior. Specifically, it shows that nicotine can shift motivation toward seeking high-reward opportunities. There are many implications for this toward human nicotine use and this will make a positive impact on the field of nicotine-related research.

That said, the authors did an excellent job addressed the issues raised by all three of the previous reviewers. Specifically, this version has improved clarity of the methodology and data presentation. Additionally, the discussion of the results and their potential impact have been significantly improved. Finally, additional data presented here (and in the supplement) support the technical premise of the study and strengthen the significance of the authors' work. Thus, I have no other substantive comments or critiques for this current version.

Reviewer #3 (Remarks to the Author):

I remain unpersuaded by the authors' rebuttal. Most notably, I don't really think that the controls or the control analyses do a very good job of disambiguating the different causes of behavior in what is a very psychologically complex task. More generally, I feel that explore/exploit tasks are not particularly novel - instead, they are a major fad - nor have they been particularly illuminating for neuroscience or psychology. Finally, I am skeptical that mouse behavior in these tasks tells us much about humans, since I suspect entirely different complex cognitive strategies are used by the two species. Having said all that, I do think the authors have made a good faith effort to respond, and I also acknowledge that the other two reviewers are much more positive than I am. As a consequence, I do think the paper should be published, on the chance that I am just a cynical curmudgeon.